# Self-interaction of NPM1 modulates multiple mechanisms of liquid–liquid phase separation

Diana M. Mitrea[1], Jaclyn A. Cika[1,2,7], Christopher B. Stanley [3], Amanda Nourse[1,4], Paulo L. Onuchic[5], Priya R. Banerjee [5,8], Aaron H. Phillips[1], Cheon-Gil Park[1], Ashok A. Deniz[5] & Richard W. Kriwacki [1,6]

Nucleophosmin (NPM1) is an abundant, oligomeric protein in the granular component of the nucleolus with roles in ribosome biogenesis. Pentameric NPM1 undergoes liquid–liquid phase separation (LLPS) via heterotypic interactions with nucleolar components, including ribosomal RNA (rRNA) and proteins which display multivalent arginine-rich linear motifs (R-motifs), and is integral to the liquid-like nucleolar matrix. Here we show that NPM1 can also undergo LLPS via homotypic interactions between its polyampholytic intrinsically disordered regions, a mechanism that opposes LLPS via heterotypic interactions. Using a combination of biophysical techniques, including confocal microscopy, SAXS, analytical ultracentrifugation, and single-molecule fluorescence, we describe how conformational changes within NPM1 control valency and switching between the different LLPS mechanisms. We propose that this newly discovered interplay between multiple LLPS mechanisms may influence the direction of vectorial pre-ribosomal particle assembly within, and exit from the nucleolus as part of the ribosome biogenesis process.

[1] Department of Structural Biology, St. Jude Children's Research Hospital, Memphis, TN 38105, USA. [2] Integrative Biomedical Sciences Program, University of Tennessee Health Sciences Center, Memphis, TN 38163, USA. [3] Biology and Biomedical Sciences Group, Biology and Soft Matter Division, Oak Ridge National Laboratory, Oak Ridge, TN 37830, USA. [4] Molecular Interaction Analysis Shared Resource, St. Jude Children's Research Hospital, Memphis, TN 38105, USA. [5] Department of Integrative Structural and Computational Biology, The Scripps Research Institute, La Jolla, CA 92037, USA. [6] Department of Microbiology, Immunology and Biochemistry, University of Tennessee Health Sciences Center, Memphis, TN 38163, USA. [7]Present address: Department of Biochemistry and Molecular Pharmacology, NYU Langone Medical Center, New York, NY 10016, USA. [8]Present address: Department of Physics, University of Buffalo, Buffalo, NY 14260, USA. Correspondence and requests for materials should be addressed to R.W.K. (email: richard.kriwacki@stjude.org)

Membrane bilayers partition cells into a variety of compartments and organelles, including the cytoplasm, nucleus, endoplasmic reticulum, Golgi apparatus, and mitochondria, which perform specialized biological functions. Cells are further compartmentalized through formation of membrane-less organelles (MLOs) such as P bodies and stress granules in the cytoplasm and nucleoli, Cajal bodies and nuclear speckles in the nucleus[1–3]. Specific sets of macromolecules self-assemble to form MLOs through a process termed phase separation, creating highly dynamic, condensed partitions with specialized functions[1]. Proteins and/or ribonucleic acids (RNAs) termed scaffold molecules[4] drive phase separation into dense liquids or hydrogels that create compositionally variable microenvironments that enable spatial and temporal control of biochemical processes[2]. The resulting condensed scaffolds recruit client[4] proteins and nucleic acids to modulate the biochemical processes localized within these MLOs.

Temporal and spatial organization of protein, deoxyribonucleic acid (DNA), and RNA macromolecules within the nucleolus orchestrate complex biological processes, including ribosome biogenesis[5,6], stress signal integration[6,7], and regulation of gene transcription[8,9]. Spatial organization is achieved through sub-compartmentalization into coexisting, immiscible liquid phases[10]. Transcription of pre-ribosomal RNA at active nucleolar organizer regions (NORs)[11,12] lowers the critical concentration for phase separation of nucleolar proteins, such as Fibrillarin and Nucleophosmin (NPM1)[10,13,14], and nucleates[11] assembly of the dense fibrillar component (DFC, Fibrillarin-rich) and, subsequently, the surrounding granular component region (GC, NPM1-rich). The immiscibility and spatial separation of these nucleolar sub-compartments is dictated by differences in their viscoelastic properties, especially in the surface tensions of the Fibrillarin- and NPM1-rich phases with respect to the surrounding nucleoplasm[10].

Genetic ablation or mRNA knock-down of nucleolar protein markers, such as Ki-67[15], NPM1[16,17], nucleolin[18], and fibrillarin[19,20], alter nuclear and/or nucleolar morphology and disrupt ribosome biogenesis. Interestingly, though, depletion of any one of these aforementioned proteins does not prevent assembly of the nucleolus, suggesting that multiple scaffolding mechanisms underlie nucleolar assembly and possibly other membrane-less bodies. We previously demonstrated that localization of NPM1 within nucleoli in live cells depends upon its ability to independently undergo liquid–liquid phase separation (LLPS) with (1) nucleolar proteins displaying multivalent arginine-rich linear motifs (R-motifs) and (2) ribosomal RNA (rRNA). The first of these mechanisms is driven by interactions between acidic tracts (A-tracts) in the N-terminal oligomerization domain (OD; A1-tract)[13,21] and intrinsically disordered region (IDR; A2- and A3-tracts)[13] of NPM1 and multivalent R-motifs within partner proteins, while the second relies upon binding of the C-terminal nucleic acid-binding domain (CTD) and the adjacent basic, disordered segment (B2-tract) of NPM1 to rRNA[13,22–24]. Figure 1a, b illustrates the clustered charges and the sub-domain organization in NPM1. Either of these heterotypic mechanisms support LLPS in vitro. In this work we hypothesized that these two mechanisms are not mutually exclusive, but operate simultaneously within the nucleolar matrix. We further hypothesized that interactions between acidic- and basic-tracts within the IDR (NPM1[IDR]) can modulate their accessibility for inter-molecular interactions with phase separation partners. We tested these hypotheses using in vitro phase separation assays as well as a variety of structural methods with wild-type NPM1 (NPM1[WT]) and mutants that either lacked critical domains/regions or had mutations within them. Our results show that, in fact, intra-IDR interactions do modulate LLPS by both heterotypic mechanisms,

and, unexpectedly, that inter-IDR interactions mediate homotypic LLPS by NPM1 under conditions that reflect physiological crowding[25]. The model that emerges is that NPM1 contributes to the liquid-like features of the GC region of the nucleolus through both heterotypic and homotypic LLPS mechanisms. This multiplicity of LLPS mechanisms may enable NPM1 to "buffer" the liquid-like features of the nucleolus as the process of ribosomal subunit assembly progresses, altering the accessibility of ribosomal proteins and RNA for interactions with NPM1.

## Results

**NPM1 and SURF6-N form heterotypic liquid-like droplets**. NPM1 is a multifunctional protein, which interacts with over 130 proteins[26], many of them annotated to be localized within the nucleolus and shown to display multivalent R-motifs[13]. One of these, Surfeit locus protein 6 (SURF6), is a non-ribosomal protein that co-localizes with NPM1 in the granular component of the nucleolus; genetic deletion of SURF6 disrupts ribosome biogenesis[27] and reduces cell viability[28]. While the specific role of SURF6 in the nucleolus is unknown, it was proposed to serve as a scaffold within the nucleolar matrix[29], albeit prior to the discovery in 2011 that nucleoli have liquid-like features[30]. SURF6 is predicted to be disordered and displays multivalent R-motifs within its sequence (Fig. 1a and Supplementary Fig. 1d, e). A N-terminal segment of SURF6, spanning residues 1–182 (SURF6-N) exhibits the 2D NMR signature of a disordered protein (Supplementary Fig. 1c) and undergoes heterotypic LLPS with NPM1. The condensation mechanism is strongly dependent on electrostatic interactions, which can be screened by increasing the solution ionic strength (Fig. 1c). We hypothesized that electrostatic complementarity of acidic-tracts within the IDR of NPM1 and R-motif-containing basic-tracts in SURF6-N drive heterotypic LLPS. We tested this hypothesis through studies of phase separation by NPM1 deletion mutants (Fig. 1b) with SURF6-N.

**C-terminal basic segment of NPM1 influences heterotypic LLPS**. We prepared two truncation mutants, one that lacks the CTD which is required for DNA and RNA binding[22–24], NPM1[N240], and another that lacks the CTD plus the B2-tract, NPM1[N188] (Fig. 1b). While its affinity for RNA is unknown, the folded CTD binds to G-quadruplex DNA sequences with affinity in the micromolar range[22,23,31] and the disordered B2-tract enhances affinity and binding kinetics[23,31,32]. We determined the threshold concentrations for phase separation of the three NPM1 constructs with SURF6-N using turbidity assays. Surprisingly, despite conservation of the acidic-tracts within all constructs, the phase boundaries for heterotypic LLPS for the two truncation mutants with SURF6-N differed from that of NPM1[WT] (Fig. 2a–c). We observed the following: (1) amongst the three constructs, NPM1[N188] phase separated at the lowest threshold concentrations of itself and SURF6-N; (2) the co-dependency concentration profile associated with the LLPS boundary for NPM1[N188] with SURF6-N, was strikingly different compared to those of NPM1[WT] and NPM1[N240]; and (3) above 4 μM, NPM1[WT] phase separated at lower SURF6-N concentrations than did the other constructs.

The threshold concentration for phase separation for a heterotypic system comprised of multivalent proline-rich motifs (PRM) and PRM-binding Src homology domain 3 (SH3) domains scaled proportionally with changes in valency of the two macromolecules, as shown both experimentally and through mathematical modeling[33]. In our studies, the binding properties of NPM1 were varied through the deletions noted above and SURF6-N was not altered; therefore, changes in the phase separation threshold for the NPM1 constructs could be

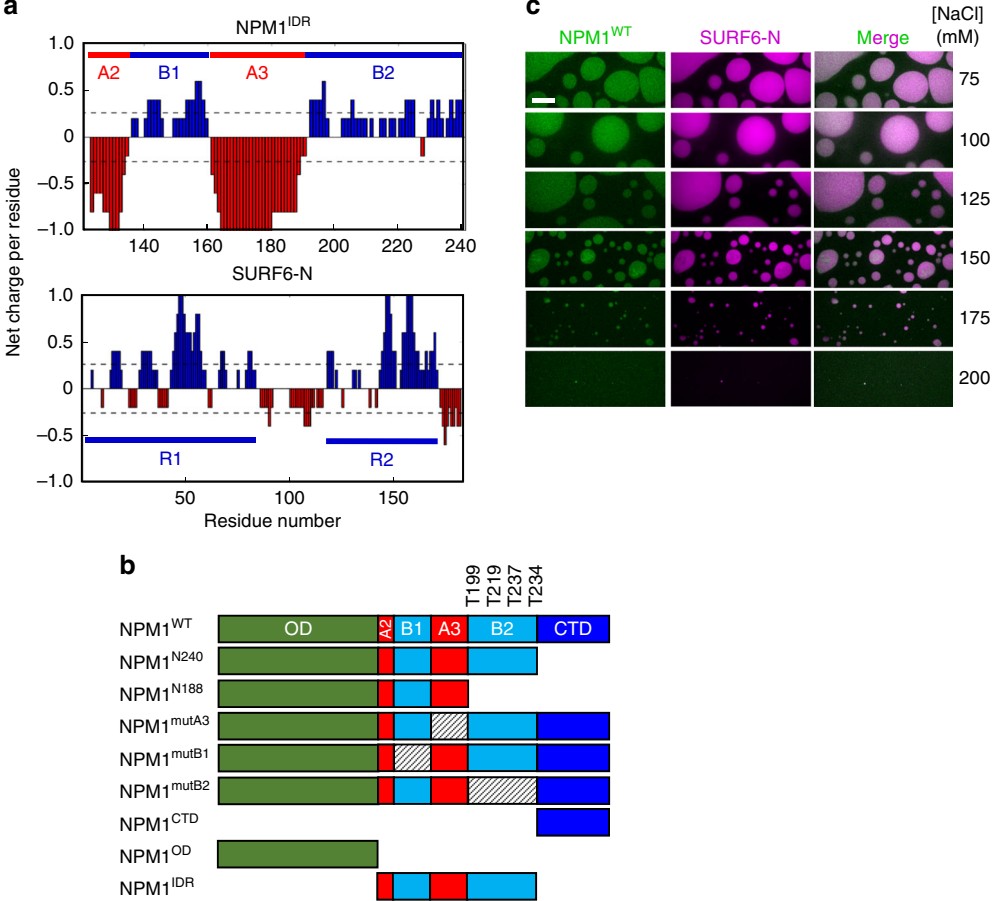

**Fig. 1** Electrostatic interactions drive NPM1:SURF6-N LLPS. **a** Net charge per residue distribution using CIDER (http://pappulab.wustl.edu/CIDER/analysis/); top: the central intrinsically disordered region (IDR) in NPM1 exhibits polyampholityc properties; IDR contains two strong acidic tracts (A2 & A3) interleaved with moderately charged basic tracts (B1 & B2); bottom: the intrinsically disordered nucleolar protein SURF6 contains multivalent R-motifs throughout its primary structure. Data are shown for residues 1–182, corresponding to the construct used in this study (SURF6-N); **b** Schematic representation of the NPM1 constructs used in this study; **c** Confocal microscopy images of phase separation by 20 μM NPM1$^{WT}$ with 20 μM SURF6-N in buffer with variable concentrations of NaCl, as indicated. Screening of electrostatic interactions at high [NaCl] disrupts LLPS droplets. NPM1 is labeled with AlexaFluor 488 and SURF6-N with AlexaFluor 647. Scale bar, 10 μm

interpreted in terms of changes in their valency for binding the R-motifs of SURF6-N. We first focused on answering the question: why does NPM1$^{N188}$ exhibit the lowest concentration threshold for heterotypic LLPS with SURF6-N compared to the other two NPM1 constructs?

**Electrostatics stabilize compact IDR conformations.** Given that the three NPM1 constructs have the same number of A-tracts, and thus the same apparent A-tract valency for interacting with R-motifs within SURF6-N, we reasoned that the differences in LLPS concentration thresholds for the constructs arise from differential accessibility of A-tracts for binding to SURF6-N. Intra[32,34]- and inter[32]-NPM1 interactions, involving the IDR, were previously shown to modulate the thermodynamic stability of the CTD, and nucleolar retention and RNA-binding properties of NPM1 in cells. We hypothesized that these interactions occur between the electrostatically complementary A- and B-tracts within IDR (Fig. 1a, b), thereby occluding the A-tracts. We further hypothesized that such intra-chain interactions between A- and B-tracts would cause compaction of the IDR, would effectively lower the valency of NPM1 for interactions with R-motifs within SURF6-N and would be disrupted by high ionic strength. To test these hypotheses, we measured the radius of gyration ($R_g$)

for the three constructs using small angle X-ray scattering (SAXS) in buffers with gradually increasing NaCl concentrations (Fig. 2d, Supplementary Fig. 2). The $R_g$ values for NPM1$^{WT}$ and NPM1$^{N240}$ exhibited positive correlation with the increasing ionic strength of the buffer (Fig. 2d). We propose that this expansion of the molecular dimensions is caused by screening of intra-IDR electrostatic interactions by NaCl. NPM1$^{N188}$, which lacks the B2-tract and CTD, does not experience [NaCl]-dependent structural expansion (Fig. 2d). The $R_g$ value measured by SAXS for this construct, as well as the pair-wise distance distribution, are constant over the entire range of salt concentrations tested (Fig. 2d and Supplementary Fig. 2c), suggesting that NPM1$^{N188}$ does not undergo ionic strength-dependent conformational changes. Molecular modeling based on SASSIE[35] (Supplementary Fig. 3), supports the hypothesis that NPM1$^{N188}$ adopts an ensemble of partially expanded conformations, likely caused by electrostatic repulsion within the highly negatively charged, truncated IDR (Fig. 1a, b; estimated charge at pH 7.5, −37.0 (http://protcalc.sourceforge.net/); Supplementary Table 1).

The loss of conformational sensitivity to ionic strength upon deletion of the B2-tract and CTD in NPM1$^{N188}$ could be explained through three distinct mechanisms which could cause IDR compaction: (1) B2-tract interacts with the A-tracts, (2) B2-tract interacts with the OD, and (3) IDR interacts with CTD. In

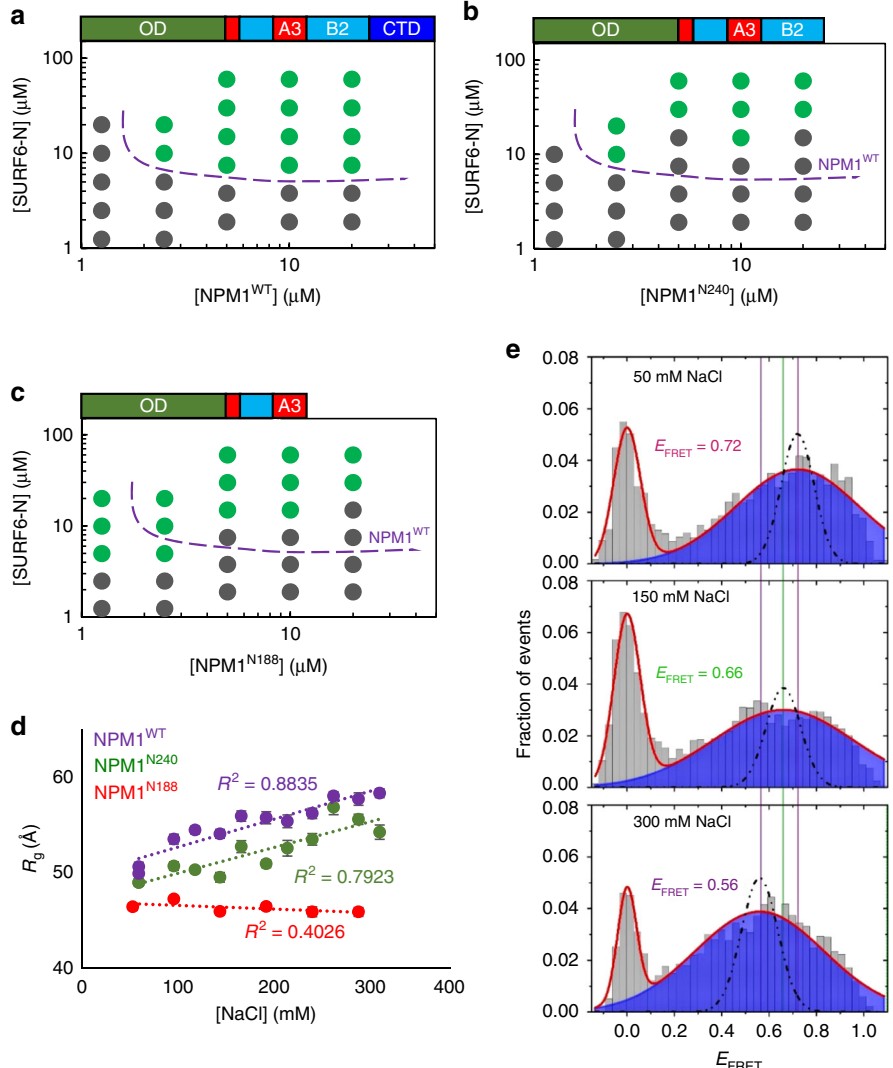

**Fig. 2** Intra-IDR interactions drive structural compaction within NPM1 and influence the threshold for heterotypic LLPS in the presence of SURF6-N. Phase diagrams determined by turbidity for NPM1$^{WT}$ (**a**), NPM1$^{N240}$ (**b**), and NPM1$^{N188}$ (**c**). The dotted purple line is a visual aid that represents the phase boundary for NPM1 between the mixed (gray circles) and demixed (green circles) states. (**d**) Changes in radii of gyration as a function of ionic strength, as determined from $P(r)$ analysis of SAXS scattering curves; fitting errors are shown. (**e**) smFRET histograms showing the variation of NPM1 conformation at increasing [NaCl]. The solid lines represent fitting of the experimental data with a Gaussian model. The peak at zero is due to molecules lacking an active acceptor dye. The dotted line indicates the shot-noise simulation at each condition

order to discriminate between these three mechanisms, we performed two-dimensional $^{1}$H/$^{15}$N HSQC (Supplementary Fig. 4a) and 1D $^{15}$N-filtered $^{1}$H diffusion (Supplementary Fig. 4b, c & Supplementary Table 3) experiments with 30 μM $^{15}$N NPM1$^{IDR}$ in the presence or absence of excess, non-isotope-labeled NPM1$^{IDR}$, NPM1$^{OD}$, and NPM1$^{CTD}$ (Fig. 1b). Small chemical shift perturbations and slowed diffusion, indicative of weak interactions, were observed for $^{15}$N NPM1$^{IDR}$ in the presence of excess NPM1$^{IDR}$, but not either of the folded domains (Supplementary Fig. 4). Thus, the NMR analysis supports a model wherein interactions between the B2-tract and the A-tracts within the IDR are responsible for the ionic strength-dependent conformational changes in NPM1$^{WT}$ and NPM1$^{N240}$.

To further test our model wherein intra-chain interactions between A-tracts and B-tracts in NPM1 cause IDR compaction and occlusion of A-tract binding sites for R-motifs within SURF6-N, we created a NPM1 construct with dual cysteine (Cys) mutations within the A2-tract of the IDR and CTD (termed NPM1$^{C125/275}$), respectively, and probed the distances between

them using dual fluorescent dye labeling (AlexaFluor 488 and AlexaFluor 594) and single-molecule Förster resonance energy transfer (smFRET). With increasing concentration of NaCl, we observed a steady shift in the FRET efficiency ($E_{FRET}$) towards lower values: 0.72 at 50 mM NaCl to 0.56 at 300 mM NaCl (Fig. 2e), suggesting a net expansion of the polypeptide chain. Furthermore, fitting of the NPM1 smFRET histograms using a Gaussian approximation indicates broadening beyond shot-noise statistics, revealing additional complexities that may arise due to heterogeneous conformations of the protein[36,37] (see Supplementary Fig. 5 and Supplementary Note 1 for further analysis and discussion). The NaCl concentration-dependent trend toward lower FRET efficiency is consistent with gradual loss of longer range A-tract/B-tract interactions that cause IDR compaction, consistent with the results from SAXS.

In summary, the results from SAXS, smFRET, and NMR support a model wherein electrostatic interactions between the A-tracts and B-tracts stabilize compact conformations of the NPM1 IDR under physiologically relevant ionic conditions.

**Charged tracts of NPM1 mediate self-association.** All three NPM1 constructs form liquid-like droplets via heterotypic interactions at 20 μM each of NPM1 and SURF6-N (Fig. 3a–c and Supplementary Movies 1–3). The dense protein phase of these assemblies exhibits liquid-like features based upon their ability to fuse upon coalescence (Fig. 3a–c, bottom panels), and the observation that both the NPM1 constructs and SURF6-N experience rapid recovery in fluorescence recovery after photo-bleaching (FRAP) experiments (Fig. 3d–f, Supplementary Fig. 6d). Interestingly, the partition coefficients, determined from the ratio of protein fluorescence intensity in the dense and light phases (see Methods and Supplementary Fig. 6a–c), for the NPM1 constructs and SURF6-N within the three different binary droplets, were strikingly different, indicative of differential composition of the dense phases (Fig. 3g–i). NPM1$^{N188}$:SURF6-N droplets incorporated ~7–10-fold less NPM1 protein and more SURF6-N within the dense phase, compared to the NPM1$^{N240}$ and NPM1$^{WT}$-containing droplets. Considering the results from SAXS (Fig. 2d) suggesting that there are intra-IDR interactions for NPM1$^{WT}$ and NPM1$^{N240}$, but not for NPM1$^{N188}$, we propose that droplets containing the longer NPM1 constructs recruit additional NPM1 molecules via inter-pentamer IDR–IDR inter-actions[32]. Interestingly, a marked decrease in the SURF6-N par-tition coefficient, accompanied by an increase in the NPM1$^{N240}$ partition coefficient, was observed for NPM1$^{N240}$:SURF6-N

droplets vs. droplets with the other NPM1 constructs, suggesting that the CTD tunes the affinity of NPM1$^{WT}$ for itself vs. R-motif containing binding partners.

We next examined the conformational features and inter-pentamer interactions of the NPM1 constructs using analytical ultracentrifugation (AUC). Specifically, two-dimensional analysis of sedimentation velocity AUC (SV-AUC) data revealed that the three NPM1 constructs sample conformations over a range of frictional ratios ($f/f_0$) and molecular mass values (Fig. 4). The molecular mass heterogeneity, which is greatest for NPM1$^{WT}$ and NPM1$^{N240}$, suggests weak self-association. In particular, a population (~20%) of dimers of pentamers was observed for NPM1$^{WT}$ (Fig. 4a and Supplementary Table 2). In contrast to the shape heterogeneity of the full-length and C-terminal truncation NPM1 constructs, the CTD (NPM1$^{CTD}$) is a homogeneous globular domain (Fig. 4d and Supplementary Table 2). These results demonstrate that pentameric NPM1 constructs that contain the full IDR experience both intra- and inter-IDR interactions. Consequently, the mechanisms that describe the phase diagrams for NPM1$^{N240}$:SURF6-N and NPM1$^{WT}$:SURF6-N are more complex than that for NPM1$^{N188}$:SURF6-N, and involve not only NPM1:SURF6-N, but also NPM1:NPM1 interactions. In summary, our collective data suggest that intra- and inter-molecular mechanisms govern the composition and

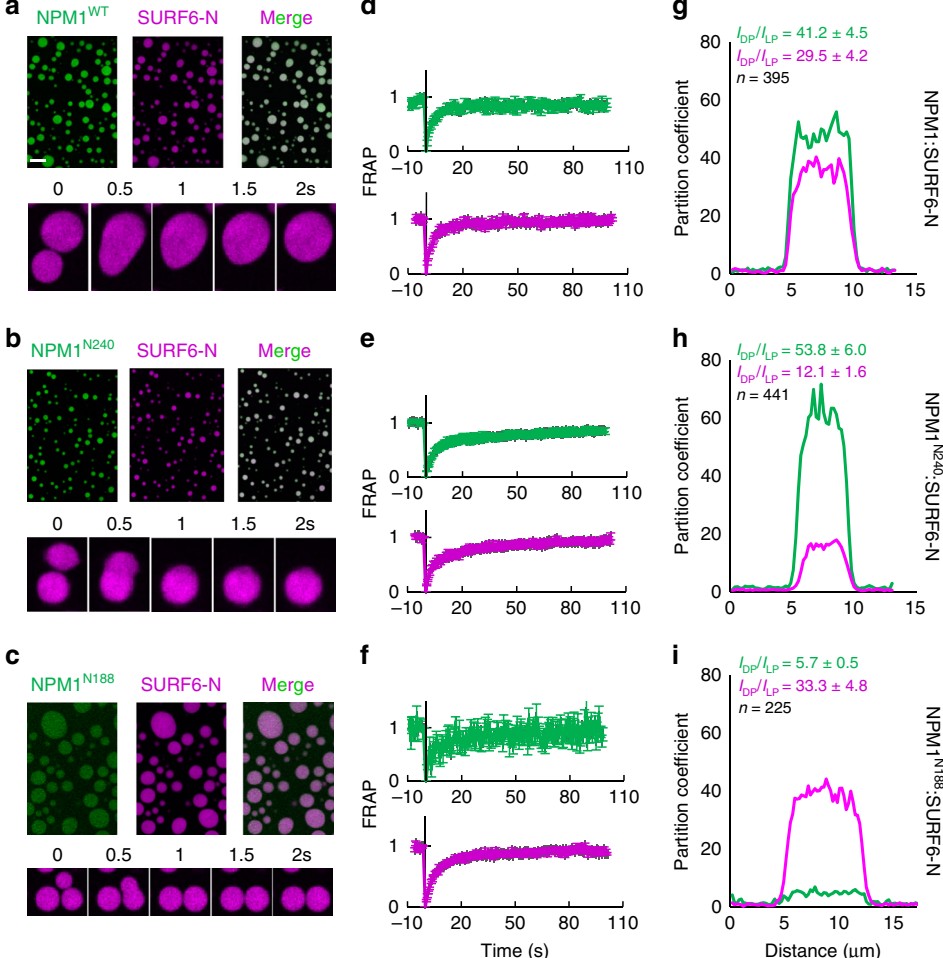

**Fig. 3** NPM1 truncation mutants form liquid-like droplets in the presence of SURF6-N of differential composition. Fluorescence confocal microscopy images (**a–c**) and FRAP curves (**d–f**) for droplets formed with 20 μM NPM1$^{WT}$ (**a**, **d**), NPM1$^{N240}$ (**b**, **e**), and NPM1$^{N188}$ (**c**, **f**) plus 20 μM SURF6-N. Representative curves illustrating cross-sections through droplets (**g–i**) illustrating differences in the partition coefficients for the NPM1 constructs and SURF6-N; values represent mean ± s.d.; $n = 5$ or as indicated on graph

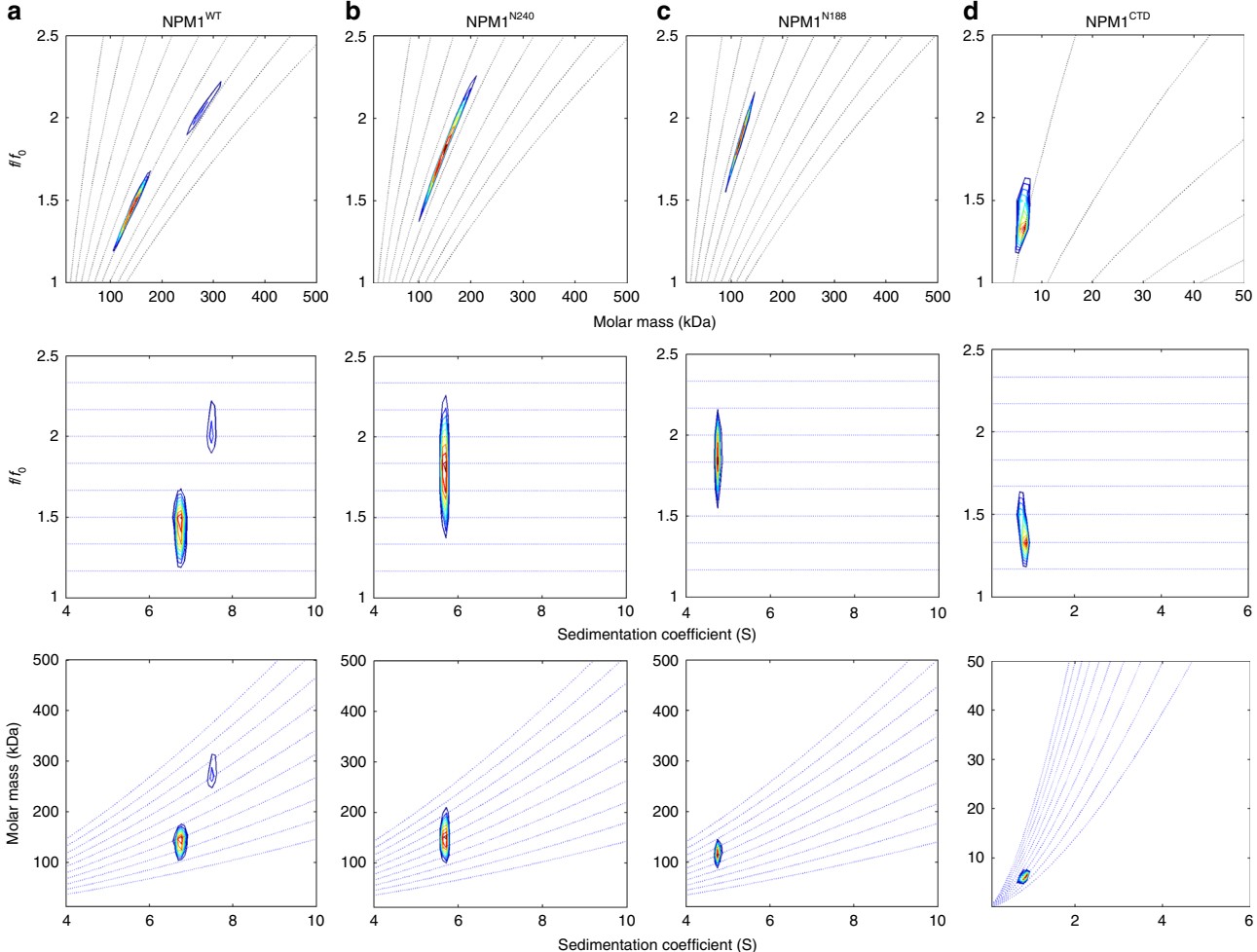

**Fig. 4** Two-dimensional size-and-shape distribution analyses of the sedimentation velocity AUC data. **a** NPM1, **b** NPM1$^{N240}$, **c** NPM1$^{N188}$, and **d** NPM1$^{CTD}$. The transformed velocity data are shown as contour plots (heat maps) of $c(M, f/f_0,)$ (top), $c(s, f/f_0)$ (middle), and $c(s, M)$ (bottom) with 0 fringes/S (white) to maximum value fringes/S (red), with increasing color temperature indicating higher values. Velocity data were acquired at 50,000 rpm at 20 °C in buffer comprised of 10 mM Tris, pH 7.5, 150 mM NaCl, 2 mM DTT. The $s_w$,$(f/f_0)_w$, and $M$-values are listed in Supplementary Table 2

network architecture of liquid-like droplets formed through LLPS of the different NPM1 constructs and SURF6-N.

**Electrostatic interactions drive homotypic LLPS by NPM1**. Self-association is known to drive homotypic phase separation of multivalent proteins[38–40]. Under conditions of physiological ionic strength, however, we did not observe homotypic LLPS of NPM1, up to concentrations of several hundred micromolar. Molecular crowding, which reaches biomolecule concentrations between 100 and 300 mg/mL in cells, can enhance polypeptide chain compaction, and protein oligomerization and aggregation[25]. In a cellular setting, the weak inter-molecular interactions between IDRs (Fig. 4 and Supplementary Fig. 4) might be stabilized, thereby inducing homotypic LLPS of NPM1. Organic polymers, such as polyethylene glycol (PEG) and Ficoll, have been used as proxies for cellular crowding[25,41,42] and have been shown to lower the concentration threshold for protein phase separation[39]. When we subjected the NPM1 constructs to crowding with 150 mg/mL Ficoll PM70, wild-type NPM1 and NPM1$^{N240}$, but not NPM1$^{N188}$, underwent homotypic phase separation at micromolar protein concentrations (Fig. 5a). Notably, the concentration threshold for homotypic LLPS was lowest for wild-type NPM1. The NPM1 constructs that exhibited homotypic LLPS also exhibited NaCl concentration-dependent compaction/

expansion (Fig. 2d, e) and high values of the NPM1 partition coefficients within the dense phase of uncrowded (e.g., lacking Ficoll PM70), heterotypic droplets with SURF6-N (Fig. 3). Together, these results support a mechanistic model wherein the NPM1$^{N188}$:SURF6-N droplets employ a pure heterotypic scaffold for phase separation involving interactions between NPM1$^{N188}$ and SURF6-N. The R-motif binding sites within the A-tracts of this construct are highly accessible and the concentration threshold for phase separation scales linearly with the component concentrations. In NPM1$^{WT}$:SURF6-N and NPM1$^{N240}$:SURF6-N droplets, in addition to the heterotypic scaffold, we propose that a homotypic scaffold also forms, resulting in increased partition coefficients for NPM1$^{WT}$ and NPM1$^{N240}$ within the dense phases (Fig. 3g–i). Compaction of the apo proteins, due to intra-IDR interactions between A- and B-tracts, lowers the valency for recruitment of R-motifs, thereby increasing the critical SURF6-N concentration threshold (for LLPS) in the low NPM1 concentration regime (<5 μM, Fig. 2a–c). At higher concentrations of NPM1 (>5 μM, Fig. 2a–c), we propose that the homotypic scaffolding mechanism becomes active, and likely competes with the heterotypic mechanism. The proposed switch in scaffold architecture explains the non-linear relationship between the NPM1 construct and SURF6-N concentrations and the concentration threshold for LLPS, as well as differences observed in the NPM1 construct partition coefficients. To develop further support for

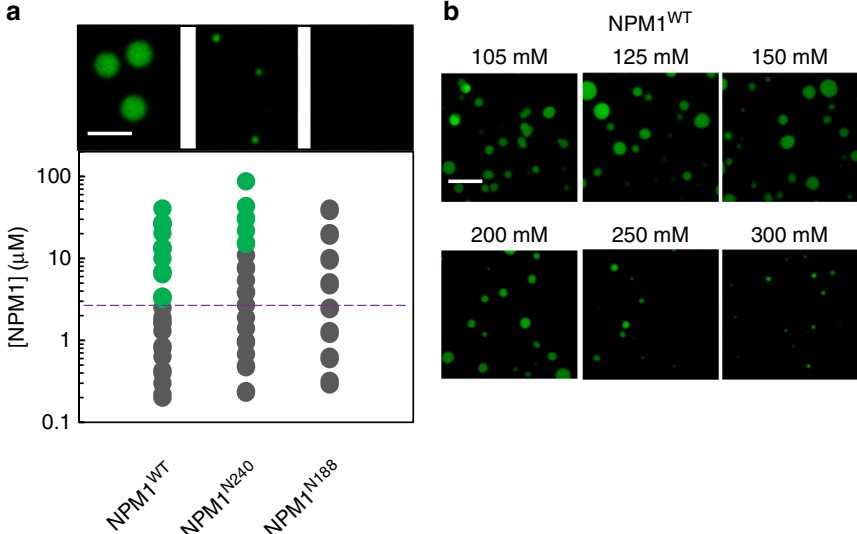

**Fig. 5** Only NPM1 constructs with an intact IDR undergo homotypic LLPS. **a** Phase diagrams for NPM1[WT], NPM1[N240] and NPM1[N188] in buffer comprised of 10 mM Tris, 150 mM NaCl, 2 mM DTT, pH 7.5 in the presence of 150 mg/mL Ficoll PM70. The purple dotted line is a visual aid that represents the phase boundary for NPM1[WT] between the mixed (gray circles) and demixed (green circles) states. Representative microscopy images are shown for 20 μM protein samples. Scale bar = 5 μm; **b** confocal microscopy images of homotypic NPM1 droplets at 20 μM NPM1 in the presence of 150 mg/mL Ficoll PM70, in buffers containing the indicated NaCl concentration. Scale bar = 10 μm

our combined heterotypic/homotypic model, we next investigated the structural features of NPM1 that are required for homotypic LLPS.

**Electrostatic A3-:B2-tract interactions drive homotypic LLPS.** Our previous studies showed that the OD and at least one disordered A-tract of NPM1 are required for heterotypic LLPS with a divalent R-motif peptide and that the OD and CTD are required for LLPS with rRNA[13]. Based on the SAXS observations showing that structural compaction of wild-type NPM1 is influenced by charge screening at high NaCl concentrations (Fig. 2d), we hypothesized that interactions between the A-tracts and B-tracts are the driving force for homotypic LLPS by NPM1. We first investigated whether homotypic LLPS is driven by electrostatic interactions. As was observed for heterotypic LLPS of NPM1 with multivalent R-motif proteins (Fig. 1c) and rRNA[10], homotypic LLPS was inhibited at high NaCl concentrations (Fig. 5b).

To further investigate the specific roles of charge patterning within the IDR in LLPS, we made NPM1 mutant constructs with mutations in the A3-, B1-, and B2-tracts. In NPM1[mutA3], we replaced the A3-tract with an equal length sequence of GGS repeats. Since the charged residues are dispersed in the B-tracts, we made Lys and Arg to Ala point mutations to create NPM1[mutB1] and NPM1[mutB2] (Fig. 1b). The protein constructs with mutations in the most highly charged tracks, namely NPM1[mutA3] and NPM1[mutB2], were unable to undergo homotypic LLPS in the concentration range tested, while NPM1[mutB1] exhibited a modest increase in the threshold concentration for homotypic LLPS (Fig. 6a). Together, these results indicate that the homotypic LLPS mechanism is primarily driven by interactions between A3- and B2-tracts. Thus, the homotypic LLPS and the structural compaction mechanisms involve interactions between disordered regions of NPM1 that are also involved in heterotypic LLPS with multivalent R-motif proteins and rRNA. Consequently, the homotypic mechanism must antagonize one or both heterotypic mechanisms. The competition for binding the A3- and/or B2-tract(s) in order to sustain one of the three types of LLPS scaffolds may act as an allosteric regulatory mechanism to control nucleolar composition based on the availability of interaction partners. Next, we tested the effect of A- and B-tract mutations on the two heterotypic LLPS mechanisms.

**Inter A3- and B2-tract interactions tune heterotypic LLPS.** We performed turbidity assays to determine the critical concentration threshold for heterotypic LLPS with SURF6-N (Fig. 6b) and rRNA (Fig. 6c) for the three IDR charge mutants. As expected, mutations in the A3- and B2-tracts severely affected LLPS with SURF6-N and rRNA, respectively, via the two different heterotypic scaffolding mechanisms (e.g., A-tracts of NPM1 interacting with R-motifs of SURF6-N, and the B2-tract & CTD of NPM1 interacting with rRNA). Interestingly, these mutations had effects not only on the respective primary LLPS mechanism, but also on the other heterotypic scaffolding mechanisms, as follows. NPM1[mutA3] exhibited a ~ten fold decrease in the threshold concentration for LLPS with rRNA (Fig. 6c), supporting the mechanistic model wherein binding of the B2-tract (to rRNA) is partially inhibited by its interaction with the A3-tract. We initially predicted that NPM1[mutB2] would similarly enhance heterotypic LLPS in the presence of SURF6-N, based on the model wherein the inhibitory interaction between the B2- and A3-tracts is removed. Contrary to our expectations, however, NPM1[mutB2] exhibited an increase in the concentration threshold for LLPS. As discussed above and shown in Figs. 2 and 3, both heterotypic and homotypic scaffolding mechanisms contribute to formation of NPM1[WT]:SURF6-N droplets, and the concentration threshold reflects convolution of the two LLPS mechanisms. We propose that the increase observed in the concentration threshold for LLPS of NPM1[mutB2] with SURF6-N (from 15 μM for NPM1[WT] to 60 μM for NPM1[mB2]; Fig. 6b) is due, primarily, to disruption of the homotypic mechanism. For comparison, the threshold concentration for NPM1[N188], for which the homotypic LLPS mechanism is also abrogated, is 30 μM (Fig. 2c). The slightly higher concentration threshold for NPM1[mB2]:SURF6-N LLPS likely arises from steric and/or electrostatic inhibition of acidic tract valency. Together, these results suggest that intra- and inter-NPM1 interactions, mediated mainly by the A3- and B2-tracts, tune the LLPS concentration thresholds for both heterotypic mechanisms.

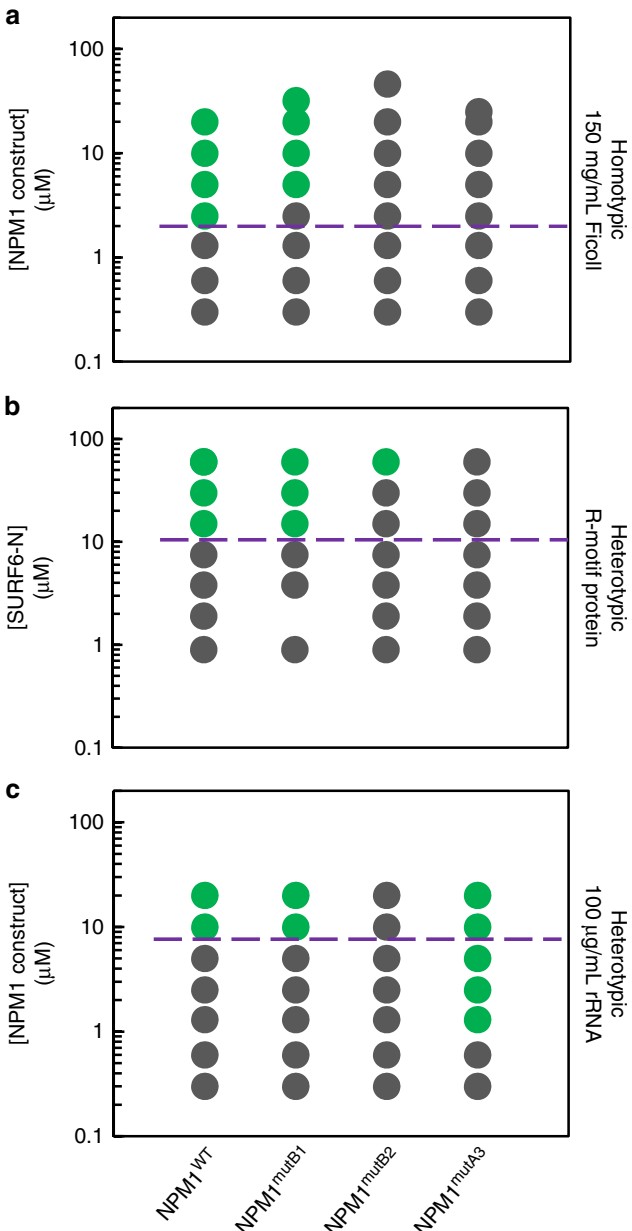

**Fig. 6** Electrostatic interactions that drive conformational compaction allosterically couple the R-motif binding and rRNA binding modes of NPM1. Phase separation diagrams based on turbidity assays for the homotypic (**a**), heterotypic with SURF6-N (NPM1 constructs at 20 μM) (**b**) and heterotypic with rRNA (**c**) mechanisms, at the indicated NPM1 construct (**a**, **c**) or SURF6-N (**b**) concentrations. The purple dotted line is a visual aid that represents the phase boundary for NPM1$^{WT}$ between the mixed (gray circles) and demixed (green circles) states

## Discussion

Here we show that NPM1 promotes LLPS through three distinct mechanisms: two heterotypic and one homotypic (Fig. 7). All three mechanisms are driven by electrostatic interactions, as demonstrated by their inhibition by high ionic strength[10] (Figs. 1c and 5b). Interactions between negatively charged A-tracts on NPM1 and positively charged R-motifs on proteins, and between positively charged B-tracts and folded CTD on NPM1 and rRNA promote LLPS through the two heterotypic mechanisms. These two heterotypic mechanisms engage different regions of NPM1 and thus are mutually compatible, allowing incorporation of two

major classes of nucleolar macromolecules—rRNA and nucleolar proteins (i.e., ribosomal and non-ribosomal)—within the same multicomponent, liquid-like matrix. Intriguingly, the homotypic mechanism relies on interactions between the A- and B-tracts, suggesting that it antagonizes both heterotypic mechanisms. What is the possible relevance of interplay between NPM1's homotypic and heterotypic LLPS mechanisms to the major function of the nucleolus, ribosome biogenesis?

Ribosome biogenesis is a complex and dynamic process. From the initial step of pre-rRNA transcription at the boundary between fibrillar centers and DFC, to the translocation of assembled pre-ribosomal particles from the nucleolus into the nucleoplasm, rRNA undergoes step-wise modifications, including splicing, posttranscriptional modifications and folding upon binding to ribosomal proteins. We hypothesize that the mechanistic redundancy associated with NPM1-dependent LLPS provides a means to maintain the liquid-like state of the nucleolar GC, while allowing vectorial assembly of ribosomal proteins with rRNA and diffusion of ribosomal subunits out of the nucleolar matrix. Specifically, we propose a model wherein different blends of NPM1's LLPS mechanisms are utilized as ribosome assembly occurs from the inside to outside of the nucleolus, as follows: (1) the heterotypic NPM1 LLPS mechanisms are dominant near the DFC, (2) rRNA and ribosomal proteins, integrated within this multicomponent scaffold, assemble into pre-ribosomal particles, sequestering sites that previously interacted with NPM1 and reducing affinity for the nucleolar scaffold, and (3) the homotypic NPM1 scaffolding mechanism takes over and becomes dominant, as the assembled pre-ribosomal particles exit the nucleolus.

Our model is supported by the following observations. First, the homotypic mechanism is promoted under conditions of molecular crowding; in our experiments, we used the branched polymer, Ficoll PM70, as a crowding agent (Figs. 5 and 6a). Linear and branched organic polymers have been shown to be suitable mimics of cellular crowding and often reproduce cellular protein folding and dynamics under in vitro conditions[41,43]. Furthermore, crowding promotes polymer chain compaction and quinary interactions[41,43]. Thus, the high local crowding within the nucleolus, wherein the components are estimated to be present at ~200 mg/mL[44], could promote both IDR compaction and non-covalent NPM1–NPM1 crosslinking by promoting intra- and inter-molecular interactions between A- and B-tracts. Ultimately, as NPM1 relies more and more on the homotypic mechanism toward the GC periphery, the mechanism may be insufficient to maintain LLPS, thus defining the boundary with the two-fold less crowded nucleoplasm (~100 mg/mL[44]). We further speculate that variations, from inside to outside, in the components and types of inter-component crosslinks that maintain the demixed nucleolar GC matrix alter the local solubility and diffusion rates of assembling ribosomal particles, possibly facilitating their vectorial transport from the site of pre-rRNA synthesis to the nucleoplasm. In our model, the affinity of the pre-ribosomal particles for NPM1 within the nucleolar matrix is low, facilitating outward diffusion and providing a mechanism for particles to exit the nucleolus.

In addition to the aforementioned posttranscriptional modifications experienced by pre-rRNA, the physico-chemical properties of nucleolar proteins are also dynamically modulated through posttranslational modifications[6,45–47]. Specifically, multiple sites within the IDR of NPM1 are known to be phosphorylated in cells[48] (Fig. 1b). The IDR participates in all three types of electrostatically-driven mechanisms of LLPS; these mechanisms could be significantly altered through phosphorylation of Ser and/or Thr residues within the IDR. For example, during mitosis, the nucleolus disassembles and residues Thr 199, Thr 219, Thr 234, and Thr 237 in NPM1, all located within the

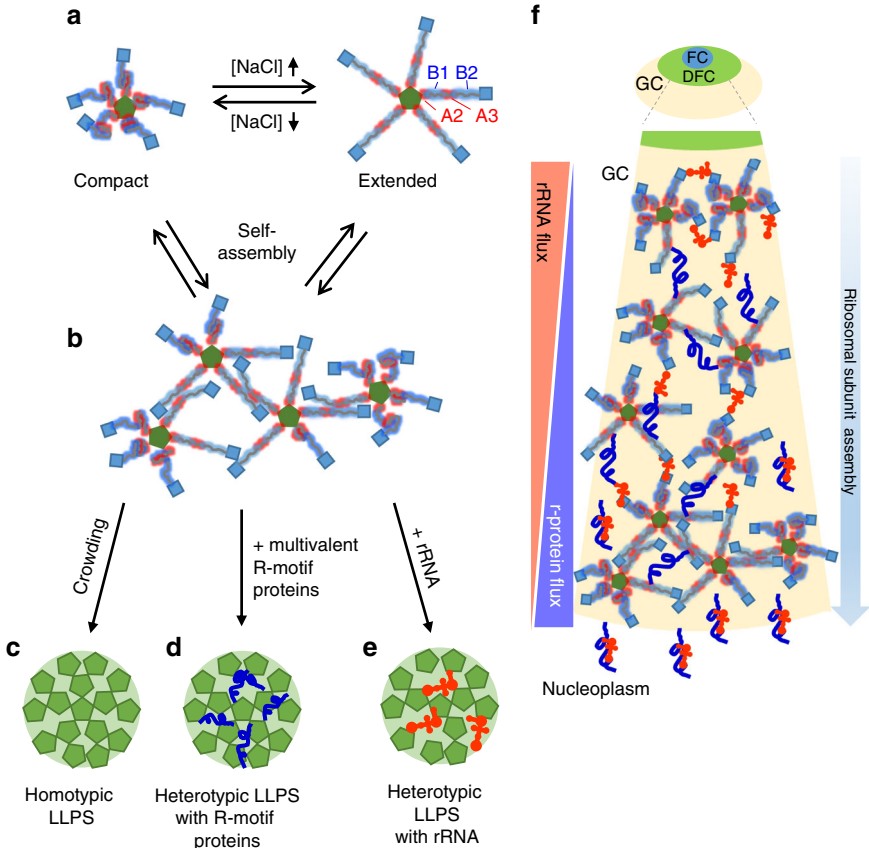

**Fig. 7** NPM1 adopts a broad range of configurations to achieve biological multifunctionality. **a** Depending upon ionic conditions and available partners, NPM1 can adopt compact conformations, through intra-chain interactions between A- and B-tracts within its IDR and, extended conformations, and can assemble into multimers through in trans interactions between A- and B-tracts. **b** NPM1–NPM1 in trans interactions form the scaffold for homotypic LLPS. In **a**, **b**, and **f**, the NPM1 OD is represented as a green pentagon, the IDR as red and blue fuzzy lines representing the A- and B-tracts, respectively, and CTD as blue squares. Homotypic interactions (**c**) can be replaced by heterotypic interactions with multivalent R-motif proteins, such as SURF6 (curvy blue line) (**d**) or with rRNA (orange objects) (**e**), changing the nature of the scaffold. Note that, for visual clarity, the entire NPM1 pentamer is represented as a green pentagon in **c**–**e**. **f** Schematic illustration of NPM1-mediated ribosomal subunit assembly in the GC of the nucleolus. Ribosomal subunits are comprised of rRNA and ribosomal proteins (r-proteins; curvy blue lines). rRNA is synthesized at the boundary between the fibrillar center (FC) and dense fibrillar component (DFC) and diffuses from the center to the periphery of the nucleolus, while the ribosomal proteins diffuse from the nucleoplasm into the nucleolus. In this model, the type of LLPS mechanism utilized by NPM1 within the GC depends upon the availability of its binding partners: rRNA, which moves outwards from the DFC/GC boundary, and R-motif proteins, which move inwards from the GC/nucleoplasm boundary, each drive different heterotypic LLPS mechanisms, in the central region of the GC. The close spatial proximity of rRNA and r-proteins within this central GC region, which exists through mixed heterotypic NPM1-mediated LLPS, enables the "hand-off" of these ribosomal components to interact with each other to form nascent ribosomal subunits (represented as conjoined orange objects and curvy blue lines). As ribosomal subunits assemble, their components no longer interact extensively with NPM1; at the same time, however, NPM1's propensity for self-interaction compensates, maintaining the liquid-like GC scaffold through homotypic LLPS

B2-tract, are phosphorylated by CDK1[47]. Phosphomimetic mutations at these positions increased the mobility of NPM1 in nucleolar FRAP assays and decreased partitioning within the nucleolus vs. nucleoplasm[47]. These phosphomimetic mutations change the net charge per residue of the wild-type NPM1 B2-tract from +0.212 to +0.135 (based on calculations of the wild-type and mutant NPM1 B2-tract using CIDER[49]). We propose that phosphorylation of these four Thr residues counterbalances the basic character of the B2-tract and inhibits its interactions with rRNA (and the heterotypic LLPS mechanism with rRNA) and with the A3-tract (and the homotypic LLPS mechanism). This illustrates how oppositely charged tracts within an IDR (A- and B-tracts) contribute directly to protein function (phase separation in the case of NPM1) and how this function may be regulated during mitosis through posttranslational modifications within one of these tracts.

The nucleolus is a remarkably complex organelle, wherein hundreds of ribosomal and non-ribosomal proteins[6] co-localize within a complex and dynamic macromolecular network supported by weak, multivalent and redundant interactions with ribosomal and other RNAs[50]. We are just beginning to understand the scaffolding mechanisms that underlie the liquid-like structure and dynamics of this membrane-less organelle. While our studies provide mechanistic insight into how NPM1 independently mediates LLPS in vitro with rRNA and R-motif-containing proteins, we acknowledge that these functions are non-essential because knockdown or deletion of NPM1 does not abrogate the assembly of the nucleolus or ribosome biogenesis[51,52]. These observations suggest that additional, redundant proteins and mechanisms contribute to demixing of nucleolar components. Clearly, further studies are needed to understand the full scope of the molecular mechanisms that create the nucleolar matrix, control its dynamics, and regulate the biological

processes that occur within it. We suggest that, in contrast to lower organisms, such as bacteria, that lack nucleoli and whose ribosomal components spontaneously assemble in vitro[53], eukaryotic cells utilize proteins like NPM1[13] and fibrillarin[10] to promote LLPS with ribosomal components not as an essential means to promote ribosome assembly, but rather to sequester these components, and essential co-factors[6], away from the nuclear milieu, allowing regulation of ribosome biogenesis and other processes within the confines of the nucleolar matrix.

## Methods

**Cloning**. NPM1, NPM1 truncation constructs and A- and B-tract mutants, and SURF6-N were cloned in a pET28a vector (Novagen) in frame with an N-terminal 6× poly-histidine tag and a TEV recognition site. The NPM1 cysteine mutants used in the smFRET experiments were cloned in a pGEX-6p-3, in frame with an N-terminal GST tag and a PreScission protease recognition site. SURF6, NPM1$^{mutA3}$, NPM1$^{mutB1}$, and NPM1$^{mutB2}$ constructs were cloned into the pET28 vector from whole gene blocks synthesized at IDT DNA. The primers used for cloning are listed in Supplementary Table 4.

**Protein expression and purification**. All recombinant NPM1 proteins were expressed in *E. coli* BL21 (DE3) strain (Novagen) in Luria Broth (RPI, Prospect, IL) media in the presence of 50 mg/L Kanamycin for His-tagged constructs or 100 mg/L Ampicillin for GST-tagged constructs. The bacterial cultures were grown at 37 °C to an optical density OD$_{600}$ ∼ 0.8. Protein expression was induced by the addition of 100 mg/L IPTG (GoldBio, St. Louis). The temperature was lowered to 20 °C and the cultures incubated overnight. Bacterial cultures were harvested by centrifugation and lysed in buffer A (25 mM Tris, 300 mM NaCl, 5 mM β-mercaptoethanol (BME), pH 7.5), by sonication on ice. The proteins were purified from the soluble fraction[13], by passing through a Ni-NTA or GST affinity column and eluting with a linear gradient of buffer A containing 500 mM Imidazole or a step gradient of buffer A containing 10 mM reduced glutathione, respectively. The affinity tags were removed in an overnight dialysis step at 4 °C, against 4 L 10 mM Tris pH 7.5, 150 mM NaCl, 2 mM DTT buffer, in the presence of TEV or HRV3C (BioVision, Milpitas, CA) protease. The cleaved proteins were loaded on a C4 HPLC column in 0.1% TFA in water and eluted with a linear gradient of 0.1% TFA in acetonitrile; the fractions containing the protein of interest were lyophilized. Lyophilized protein was resuspended in buffer containing 6 M guanidinium hydrochloride to a final monomer concentration of 100 μM and refolded by dialysis against three changes of 1 L 10 mM Tris, 150 mM NaCl, 2 mM DTT, pH 7.5, at 4 °C.

SURF6-N protein was expressed in BL21 Rosetta (DE3) strain, with a 3 h incubation at 37 °C post IPTG induction. Bacterial pellets were harvested by centrifugation and SURF6-N was purified from the insoluble fraction. SURF6-N was solubilized in 25 mM Tris pH 7.5, 6 M guanidinium hydrochloride, 5 mM BME buffer and loaded on a Ni-NTA affinity column, and eluted with a linear gradient of the loading buffer, containing 500 mM imidazole. The chaotropic agent was removed by overnight dialysis and the poly-histidine tag was removed by proteolytic cleavage with TEV. The cleaved protein was further purified on a C4 HPLC column and lyophilized, using the same buffers as for NPM1 constructs. SURF6-N was reconstituted in 6 M guanidine hydrochloride buffer and dialyzed against 10 mM Tris, 300 mM NaCl, 2 mM DTT, pH 7.5 for long term storage at −80 °C. Working stocks of SURF6-N were prepared by diluting the high salt storage aliquots to <200 μM protein and a final [NaCl] of 150 mM.

**Turbidity assays**. Samples were prepared in 10 mM Tris, 150 mM NaCl, 2 mM DTT, pH 7.5 buffer as indicated, in a total volume of 10 μL, by diluting NPM1 in the buffer, followed by the addition of the component initiating the phase separation [i.e., SURF6-N, wheat germ rRNA (BioWorld, Dublin, OH), Ficoll PM70 (GE Healthcare, Marlborough, MA)]. The samples were incubated at room temperature for 15 min, and absorbance at 340 nm was read in triplicate on a NanoDrop 2000c spectrophotometer (Thermo Scientific, Waltham, MA). Each turbidity assay was reproduced in triplicate.

**Fluorescent labeling**. NPM1 constructs and SURF6-N were labeled with maleimide AlexaFluor488-C5 and maleimide AlexaFluor647-C2 (Thermo Fisher Scientific, Waltham, MA), respectively, according to the manufacturer's protocol and as described previously[13]. NPM1 constructs were labeled at Cys104 and SURF6-N was labeled at Cys19. In order to minimize potential structural and optical artifacts arising from clustering of Alexa488 dyes on the pentameric ring of NPM1$^{OD}$, we mixed 10% fluorescently labeled with 90% unlabeled NPM1 in 6 M guanidinium hydrochloride buffer and refolded by dialysis, to promote incorporation of a single labeled monomer per pentameric ring.

**Microscopy**. All images were acquired using a 3i Mariannas spinning disk confocal microscopy instrument (Intelligent Imaging Innovations, Inc., Denver, CO), using a 63× oil immersion objective, NA 1.0. Samples were prepared in 10 mM Tris, 150

mM NaCl, 2 mM DTT, pH 7.5, unless otherwise noted, and incubated at room temperature for ∼1 h before imaging in CultureWell 16-well chambered slides (Grace BioLabs, Bend, OR). The slides were coated with SigmaCote (Sigma-Aldrich, St. Louis, MO) and Pluronic F-127 (Sigma-Aldrich, St. Louis, MO). Droplets form in the bulk solution and settle due to gravity onto the surface of the coverslip, as shown in Supplementary Movies 1–3. All reported images were recorded at the coverslip surface. Due to the rapid movement of the floating droplets, imaging and subsequent quantification of them was not possible, due to them drifting out of focus. For the FRAP experiments, a circular region of interest (ROI) of 1 μm in diameter, located at the center of the droplets was photobleached to ∼50% intensity by illuminating the ROI with the laser set at 100% power for 1 ms.

**Fluorescence spectroscopy**. Aqueous stock solutions of Maleimide AlexaFluor 488-C5 and maleimide AlexaFluor 647-C2 fluorescent dyes were diluted 1:50 to 1 μM into water:glycerol solutions at the specified percentage glycerol. Emission spectra were collected at 20 °C, using a QuantaMaster400 fluorometer (Horiba, Kyoto, Japan). AlexaFluor 488 samples were excited at 485 nm and emission data was recorded between 490 and 600 nm. AlexaFluor 647 samples were excited at 645 nm and emission data recorded between 650 and 800 nm (Supplementary Fig. 6a, b). In order to correlate the quantum yield changes observed by fluorescence spectroscopy with the microscopy data, we integrated the emission intensity measured spectroscopically over the wavelength range of the microscope bandpass filters: 525/50 BP (500–550 nm) and 725/150 (650–800 nm) for AlexaFluor 488 and AlexaFluor647, respectively. The integrated emission intensities were normalized with respect to the value for each AlexaFluor dye in water. Microrheology measurements of viscosity values within liquid-like and gel-like protein droplets indicated values between 0.7 Pa·s and 100 Pa·s[10,38,54]. Therefore, we defined the correction factor for each dye as the average of relative change values in integrated emission signal at $\eta > 0.3$ Pa·s, where the variation appears to plateau (Supplementary Fig. 6c).

**Image analysis to determine partition coefficients**. Images were acquired and processed in SlideBook 6.0 software (Intelligent Imaging Innovations, Inc., Denver, CO). The partition coefficient was defined as the intensity within the dense phase ($I_{DP}$) divided by the intensity in the light phase ($I_{LP}$). Three images per condition were analyzed and the total number of objects quantified is shown in Fig. 3g–i. Specifically, we defined the intensity of the light phase, in each image analyzed, as the mean intensity within a circular region of interest (ROI) with a diameter of 5 μm, placed in the background, away from any fluorescent objects (i.e., droplets). Next, we defined masks based on the intensity in the 640 nm channel (SURF6-N fluorescence signal). The SURF6-N intensity was chosen over that of NPM1-construct due to its higher signal to noise ratio across all three samples. The masks were defined around objects with a diameter larger than 2 μm, using the Otsu threshold method[55]. $I_{DP}$ represents the mean intensity under the mask for each individual channel. The partition coefficients were calculated as $[(I_{DP}-I_{bkrd})/c_{corr}]/(I_{LP}-I_{bkrd})$, where $I_{bkrd}$ represents the mean intensity of an ROI in the center of an image of buffer alone and $c_{corr}$ is the quantum yield correction coefficient for each dye (0.73 for AlexaFluor 488 and 1.43 for AlexaFluor 647).

**Small angle X-ray scattering**. SAXS experiments were conducted using a Rigaku BioSAXS-2000 home source system with a Pilatus 100K detector and a HF007 copper rotating anode (Rigaku Americas, The Woodlands, TX). Data were collected at a fixed sample-to-detector distance using a silver behenate calibration standard. The instrument software was used to reduce the data to scattering intensity, $I(Q)$, vs. wave vector transfer, $Q$ ( $= 4\pi \sin(\theta)/\lambda$, where $2\theta$ is the scattering angle), and then subtract the buffer background. The scattering curves were collected at a protein concentration of 30 μM, at 25 °C, for 1 h. The reduced data were analyzed with the ATSAS suite[56], using PRIMUS and GNOM for Guinier and $P(r)$ analyses, with a $q$ range between 0.018 and 0.31 Å$^{-1}$. Reported $R_g$ values were obtained from the $P(r)$ analysis performed with GNOM[56], although similar values were obtained using the Guinier approximation performed with Primus[57].

**SASSIE**. The SASSIE program[35] was used to generate ensemble models for NPM1$^{N188}$ to compare with the SAXS data. Using the pentameric core structure in PDB ID 4N8M as the starting model, the remaining NPM1$^{N188}$ sequence was generated for each subunit using the psfgen package within VMD/NAMD[58]. SASSIE then was used to minimize the structure and run Monte Carlo simulations with the pentamer core kept fixed (PDB ID: 4N8M[21]). The models were then compared to the SAXS $I(Q)$ curve for NPM1$^{N188}$ (150 mM NaCl) to obtain $\chi^2$ values. We sampled the potential range of the conformational landscape with SASSIE to create representative conformers and then compared each conformer to the experimental curve to obtain a $\chi^2$ value. The $\chi^2$ vs. $R_g$ plot (Supplementary Fig. 3) exhibits a broad basin near the experimentally observed $R_g$ value of 44 Å, suggesting that NPM1$^{N188}$ dynamically exchanges between conformers with a range of compaction values. However, because the SAXS curves reflect only the ensemble averaged conformation, we cannot define minimum and maximum compaction limits for NPM1$^{N188}$ molecules that experience conformational averaging.

**Single-molecule FRET**. The NPM1 dual-Cys construct, C125/275, was fluorescently labeled with Alexa488/Alexa594 FRET dye-pair using maleimide-functionalized dyes (Life Technologies/ThermoFisher Scientific)[13]. Single-molecule experiments were performed at 200 pM labeled NPM1[125–275] in a 20 nM excess of unlabeled wild-type NPM1 using a home-built set up[59]. All experiments were performed in 10 mM Tris buffer, pH 7.5. Samples were prepared in 50 mM, 150 mM, and 300 mM NaCl. Briefly, laser illumination was used to excite the donor dye, and photon emission from the donor and acceptor dyes was recorded continuously and simultaneously as a function of time with an integration time of 500 µs, using avalanche photodiode single-photon counting modules (SPCM-AQR-14, Perkin-Elmer; now Excelitas, Waltham, MA) and a counting card (PCI-6602, National Instruments, Austin, TX). The detection volume was placed several µm from the coverslip surface to probe freely diffusing molecules, minimizing potential surface-induced perturbation in their structural properties[60,61]. FRET efficiency ($E_{FRET}$) values were calculated ratiometrically using the intensities of the photon bursts from single molecules on the donor and acceptor channels[62]. $E_{FRET}$ data from several thousand molecules were subsequently used to plot $E_{FRET}$ histograms, which directly provide information about structural distributions, dynamics, and subpopulations. Data analysis was performed using NLS fitting algorithm of OriginPro 8.6.

**smFRET histogram simulation from shot-noise statistics**. The smFRET peak broadening due to photon shot-noise (inherent noise due to the statistics of detecting a limited number of photon counts from single molecules) was estimated[59] based on the theoretical framework by Gopich and Szabo[36]. Inherent to this model is the assumption that NPM1 conformational fluctuations, if any, are much faster than the integration time (0.5 ms) used in our experiments, i.e., corresponding to histograms for a single weighted-average protein conformation broadened only due to the Poissonian distribution of photons. The calculated FRET efficiency distribution has the form:

$$\rho_{poi}(E|\varepsilon, N) \approx \left[2\pi\sigma_{poi}^2(\varepsilon, N)\right]^{-1/2} \exp\left(-\frac{(E-\varepsilon)^2}{2\sigma_{poi}^2(\varepsilon, N)}\right) \quad (1)$$

Here, $\sigma_{poi}^2(\varepsilon, N)$ is the variance of the FRET distribution. The calculated smFRET histograms based on Equation (1) are shown in Fig. 2e and Supplementary Fig. 5b. In respective cases, the mean $E_{FRET}$ was assumed to be the same as the center of the Gaussian fitting of the experimental data, while the variance was calculated based on the respective threshold values ($N_T$). The variation of the peak width was also tested based on different $N_T$, to get an upper estimation of the smFRET peak width. This is shown in Supplementary Fig. 5b.

**Analytical ultracentrifugation**. Sedimentation velocity experiments were conducted in a ProteomeLab XL-I analytical ultracentrifuge (Beckman Coulter, Indianapolis, IN) following standard protocols unless mentioned otherwise[63]. The samples, dialyzed overnight against the reference buffer (10 mM Tris pH 7.5, 150 mM NaCl, and 2 mM DTT) were loaded into a cell assembly comprised of a double sector charcoal-filled centerpiece with a 12 mm path length and sapphire windows. Buffer density and viscosity were determined in a DMA 5000 M density meter and an AMVn automated micro-viscometer (both Anton Paar, Graz, Austria), respectively. The partial specific volumes and the molecular masses of the proteins were calculated based on their amino acid compositions in SEDFIT (https://sedfitsedphat.nibib.nih.gov/software/default.aspx). The cell assembly, containing identical sample and reference buffer volumes of 360 µL, was placed in a rotor and temperature equilibrated at rest at 20 °C for 2 h before it was accelerated from 0 to 50,000 rpm. Rayleigh interference optical data were collected continuously for 12 h. The time-corrected velocity data were analyzed[64] with SEDFIT (https://sedfitsedphat.nibib.nih.gov/software/default.aspx) using the two-dimensional size-and-shape model, $c(s, f/f_0)$ (with the one dimension the $s$-distribution and the other the $f/f_0$-distribution)[65]. Calculation was with an equidistant $f/f_0$-grid of 0.15 steps that varies from 1.0 to 2.5, a linear $s$-grid with 100 sedimentation coefficient values from 2 to 10 S for NPM1[WT], NPM1[N240], and NPM1[N188] and 0.1 to 8 S for NPM1[CTD]. Tikhonov–Phillips regularization was at one standard deviation. The velocity data were transformed to $c(M, f/f_0)$, $c(s, f/f_0)$, and $c(s, M)$, and distributions with $M$ the molar mass, $f/f_0$ the frictional ratio, and $s$ the sedimentation coefficient and plotted as color temperature contour maps. The dotted lines of $c(M, f/f_0)$ indicate constant $s$ and that of the $c(s, M)$ plot constant $f/f_0$. These distributions were not normalized[63,65].

**NMR spectroscopy**. All experiments were collected on Bruker spectrometers with cryogenically cooled probes. The 2D $^1H/^{15}N$ HSQC spectrum of SURF6-N was collected at a $^1H$ Larmor frequency of 800 MHz on a ca. 0.5 mM sample. Titrations of $^{15}N$-labeled NPM1[IDR] with unlabeled NPM1[OD], NPM1[IDR], and NPM1[CTD] were collected at 700 MHz. $^{15}N$-filtered diffusion experiments were collected at 600 MHz. Diffusion coefficients were measured by incorporating a longitudinal encode–decode element[66] before transfer of magnetization to $^{15}N$ in an HSQC experiment without incrementing the delay in the indirect dimension. Individual

1D spectra were collected with 256 scans at $z$-gradient strengths of 20, 24.5, 28.3, 31.6, 34.6, and 37.4% of the maximum strength. Data sets were measured in triplicate at each gradient strength and then the bulk intensities of the amide envelope were fit as a group to determine the diffusion constant, based on Equation (2). Errors in the diffusion constant are the standard error to the fit from all 18 data points. Diffusion curves are represented by the averaged data, error bars represent the standard deviation. The radius of hydration was determined from the diffusion constant using the Stokes–Einstein Equation (3) with errors determined by propagating the errors in the diffusion constants. Gradient strengths were calibrated to yield an $R_H$ of the NPM1[IDR] at 30 µM of 2.3 nm as determined by dynamic light scattering using a DynaPro NanoStar (Wyatt Technology, Santa Barbara, CA) instrument. The gradient strength at 100% was 0.423 T m$^{-1}$. Data were processed with Topspin (Bruker, Billerica, MA) and analyzed with CARA (http:/cara.nmr.ch).

$$\ln I/I_0 = -D\, \Upsilon^2 g^2 \delta^2 (\Delta - \delta/3) \quad (2)$$

where $I$ and $I_0$ is the observed and reference signal intensity, $D$ is the diffusion constant, $\Upsilon$ is the $^1H$ gyromagnetic ratio, $g$ is the gradient strength, $\delta$ is the length of the gradient, and $\Delta$ is the diffusion delay.

$$D = kT/6\pi\eta R_H \quad (3)$$

where $k$ is the Boltzmann constant, $T$ is the temperature, $\eta$ is the viscosity, and $R_H$ is the hydrodynamic radius.

**Data availability**. All relevant data are available from the authors; submit all requests to the corresponding author.

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

## Acknowledgements

Images were acquired at the Cell & Tissue Imaging Center which is supported by St. Jude Children's Research Hospital (SJCRH) and NCI Cancer Center Support grant P30 CA021765. The small-angle X-ray scattering experiments were supported by DOE scientific user facilities. We thank Taehyung C. Lee for assistance with smFRET studies and Dr. Mylene Ferrolino and Nazia Ahmed for assistance with protein purification. This work was supported by funding from NIH 5RO1GM115634 (to R.W.K. and A.A.D.), an NCI Cancer Center Support grant (P30 CA21765 at SJCRH), NIH RO1 GM066833 (to A. A.D.), and ALSAC.

## Author contributions

D.M.M. and R.W.K. designed the study, interpreted data and wrote the paper; D.M.M. and J.A.C. performed microscopy and turbidity assays, analyzed and interpreted data; C.B.S. and D.M.M. performed SAXS experiments, analyzed and interpreted data; A.N. performed AUC experiments, analyzed and interpreted data; P.L.O., P.R.B. and A.A.D. designed and performed smFRET experiments, analyzed and interpreted data; A.H.P.

performed NMR experiments, analyzed and interpreted data; C.-G.P. performed cloning and protein purification.

## Additional information

**Competing interests:** The authors declare no competing financial interests.

