## [Peer Review File · Nature Communications]

Reviewers' comments:

Reviewer #1 (Remarks to the Author):

Self-Interaction of NPM1 Modulates Multiple Mechanisms of Liquid-Liquid Phase Separation
Comments:

This is an interesting and very important study that add significantly to the field and will have a noticeable impact. The manuscript is well-written and concise. However, there are several issues than need to be addressed:

1) Page 3-Introduction, 3rd Paragraph, second sentence: "...depletion of any one of these proteins does not prevent assembly of the nucleolus..."

a. Any one of the proteins listed in the sentence above or any nucleolar protein? The authors mention that NPM1 interacts with over 130 proteins and that many of those are in the nucleolus and have multivalent R-motifs.

b. Has it been shown that a deletion of a combination of these proteins prevents assembly of the nucleolus?

c. If NPM1 knockdown does not prevent assembly of the nucleolus then is LLPS separation functionally crucial for processes like ribosome biogenesis or are there compensatory mechanisms?

2) Page 4-Introduction, 3rd Paragraph, eighth sentence: "...that inter-IDR interactions mediate homotypic LLPS by NPM1 under moderately crowded conditions."

a. What defines moderately crowded conditions?

b. When does this occur physiologically and what are the functional implications?

3.) Page 14-Methods, 3rd Paragraph third sentence: "...Each turbidity assays was reproduced..."

a. "assays" should be changed to "assay"

4) Figures 2/5/6: The authors used green circles to indicate, most likely conditions, when LLPS occurred. However, this is not clear and needs to be indicated in the figure legend.

5) Figure 7: This is a nice figure, but it would be ideal to have an explanation of the possible functionality of the different types of interactions to show the biological relevance.

6) Figure 3 ai/bi/ci: The scale bar is barely visible and could be enlarged.

7) The scale bar on some figures (Figure 3 ai/bi/ci, Figure 5b) are labeled with the value (10 um) but others are not (Figure 1C, Figure 5A). It would be ideal if they were all consistent.

8) There are two scale bars on Figure 1C (one vertical and one horizontal). This is not the case on any other figure and is redundant and could be removed.

April Darling and Vladimir N. Uversky

Reviewer #2 (Remarks to the Author):

The manuscript describes experiments to examine how homotypic and heterotypic interactions between different regions of NPM1 and between NPM1 and SURF6 affect phase separation of these proteins.

As noted by the authors, several proteins play role in nucleolar morphology, including Ki-67, NPM1, nucleolin and fibrillarlin, but loss of any one of these proteins does not prevent assembly of the nucleolus. This hints at the complexities of the interactions between these morphology determining proteins, a theme that the authors explore via their study of NPM1 and SURF6. NPM1 is an excellent test case because its obvious acidic tracts and basic tracts clearly define an electrostatic mechanism that is easily disrupted by increasing salt concentration.

The authors first demonstrate that NPM1 can form droplets with SURF6-N (SURF6 res 1-182)

presumably via interactions between the acidic tracts in NPM1 and arginine-motifs (R-motifs) in SURF6, and that these droplets can be disrupted by increasing ionic strength. Using truncation mutants, they further demonstrate that threshold concentrations for phase separation change depending on which portions of NPM1 are deleted, but not in a linear fashion, as might be expected for deletion of equivalent binding nodes in a multivalent system. Instead, the shortest construct used exhibits the lowest concentration threshold for LLPS with SURF6-N. SAXS and FRET data indicate that NPM1-WT and NPM1-N240 have intramolecular electrostatic interactions that lead to compaction and can be disrupted by salt. In contrast, ionic strength had minimal effect on NPM1-N188. Thus, intramolecular interactions were postulated to occlude interaction nodes that promote LLPS with SURF6-N in the longer constructs. Partition coefficients and ratios of NPM1 to SURF6-N are different for the three different NPM1 constructs supporting the notion that NPM1 binding site valency is changing. NPM1 intramolecular interactions can transition to intermolecular interactions under crowding conditions leading to LLPS for longer NPM1 constructs (in the absence of SURF6-N), but not for NPM1-N188 which has lost complementary binding sites. Thus, homotypic and heterotypic interactions are in competition in LLPS of NPM1 and SURF6-N. The electrostatic nature of the LLPS driving interactions was further examined by mutation of the acidic or basic tracts. These experiments also explored the tunability of LLPS concentration thresholds.

Overall, the results are novel, with direct implications for the mechanisms underlying formation of the nucleolus, and will definitely be of interest to many in the cell biology, biophysics and RNA communities. The data are beautifully presented and the paper is well written. The paper explores unique ideas around the role of intramolecular interaction in the tunability of LLPS, hinting at mechanisms for redundancy and robustness in the formation of the nucleolus.

Comment:

The polymer phase separation field suggests that compaction often accompanies phase separation while there have been alternative suggestions within the protein phase separation community. Figure 7 depicts self-assembly in an extended conformation. Could the authors be more pedagogical in explaining how their data support this model or if there could be alternative interpretations?

Reviewer #3 (Remarks to the Author):

A major claim of the paper is that the acidic and basic tracts in the intrinsically disordered regions, IDR, mediate the formation of liquid pure NPM1 phases and mixed NPM1 SURF6-N phases via electrostatic interactions.

The findings are novel, of interest to the community studying liquid phase transitions of biomolecules, and the wider fields of chemistry, biology, biochemistry, and biophysics.

A clear-cut presentation of the experimental data, more detailed analysis and a better documentation thereof will significantly strengthen the claims without the need for additional experiments. Given the current presentation, certain claims may seem speculative without directly supported by the experiments for non-experts.

The paper may influence the thinking of the field by stressing the importance of electrostatic interactions in the formation of liquid phases of biological macromolecules. However, neither the data analysis nor the concepts are revolutionary.

The authors may consider to seeking for more support by the collaborating experimentalist in the data analysis and documentation thereof. The data and the analysis thereof is unfortunately not fully

documented. Seemingly the document underwent several iterations to arrive at its current stage. However, the manuscript seems to be submitted without a final proofreading.

Overall, I highly appreciate the amount of work and experiments that lead to this manuscript. Therefore, I recommend a publication in Nature Communications after significant improvements of the manuscript, the data analysis, and the interpretation of the data.

Please find a more detailed review below.

General remarks

I was excited to review the highly exciting manuscript "SELF-INTERACTION OF NPM1 MODULATES MULTIPLE MECHANISMS OF LIQUID-LIQUID PHASE SEPARATION". The authors applied confocal microscopy, SAXS, analytical ultracentrifugation, and single-molecule fluorescence to study the liquid-liquid phase separation (LLPS) of nucleophosmin (NPM1). The influence of the NPM1 and salt concentration and effect of heterotypic interactions with potential binding partners, such as SURF6-N (Surfeit locus protein 6), was studied. The authors consider possible interaction within one component phases of NPM1 (homotypic) and mixed phases of NPM1 with other proteins and binding partners (heterotypic). They attempt to decipher these interactions, which makes the paper highly interesting.

Unfortunately, the presented conclusions are mainly qualitatively, and the data is analyzed very superficially. The manuscript may benefit significantly by focusing more on the information provided by the experiments. The authors may concentrate in more detail on applied techniques and the probed sample properties to clarify how they come to their scientific conclusions. Given the word count limits, detailed information could be provided by either the online methods or the supplementary material.

Finally, certain statements seem reasonable but are not proven by the experiments. The authors repetitively claim an interaction within the IDR of the A/b-tracts. Assuming such interaction is reasonable; however, their data or the analysis of thereof, i.e., the salt dependence of the R_g , cannot exclude interactions of the IDR with the folded core of the NPM1 pentamer.

Most scientific and technical remarks could be addressed, by either more detailed analysis, a better presentation thereof, or weakening of the claims.

Besides these aspects, the manuscript will require significant editorial improvements before publication. Below find a more detailed list of questions and concerns.

Experimental remarks

Microscopy) (a) It is unclear how the microscopic images were obtained. The images were probably taken on the surface of coverslips. If this is the case, does the NPM1 phase form in the liquid bulk phase or the glass water interface? (b) It seems that the fluorescence quantum yields were not accounted for in the analysis of the partition coefficients. Alexa647 and Alexa488 have significant different quantum yields, QY. Moreover, Alexa647 is an environment sensitive dye, i.e., viscosity dependent QY. If the presented analysis considered the QY and the detection efficiencies of the channels, the analysis should be described in more detail.

SAXS) (a) The experimental SAXS-curves, the model function, and the analyzed q-range should be presented in a figure including the weighted residuals of the models. (b) The analysis by SASSIE is nearly undocumented. It is unclear if the χ^2 (supplementary Figure 2) is calculated for a single structure or an ensemble of structures. (c) If the χ^2 corresponds to single structures out of an ensemble, the authors should consider weakening the statement that NPM1 adopts ensembles of partly compact states, as a mixture of extended and collapsed structures may also describe the data.

smFRET) When describing the smFRET setup the authors refer to Ferreon et al. 2009 where ALEX-excitation was used. In the manuscript solely the donor-excitation is mentioned. Did the authors employ ALEX or PIE? If this is the case, does the variance of acceptor brightness distribution explain the broad FRET efficiency histogram? Do the authors select FRET molecules by the directly excited acceptor? If so, which selection criteria were used?

FRAP) The recording of the FRAP curves is unclear. Which region of interest, ROI, was bleached? An entire NPM1 droplet or parts a droplet? Together with the FRAP curves the ROI before bleaching, after bleaching, and the recovered region could be shown.

Analytical ultracentrifugation) Figure 4 may be simplified to stress the statement that the sample is heterogeneous. The main finding is that NPM1WT forms dimers of pentamers and that NPM1N240 is heterogeneous. The supplementary table 2 seems not to coincide to Fig. 4 of the main text.

Editorial remarks

- * The abbreviation SURF6, Surfeit locus protein 6, is not introduced.
- * The naming of SURF6-N is inconsistent, in the figures SURF6-N is sometimes addressed by SN6 (Fig. 6)
- * The abbreviation A-tracts is used before being defined.
- * Figures are referred without discussion of their content, e.g., Fig. 1a.
- * The list of mutants (Fig. S1) is useful and thus may be presented in the main text Fig.1.
- * The superscripts of the header of supplementary table 2 are wrong. Please also give the protein NPM1 concentration in mM for easier comparison with the other techniques.
- * Most figures are insufficiently described by their legends, e.g., in the legend of Fig. 1c it is not mentioned that microscopic images are shown.
- * Fig. 1c. AlexaFluor647 instead of AlexaFluor640 - I guess

Scientific questions

Is it also possible, that the IDR is either bound to the folded pentameric unit of NPM1 or "loose"? Such mechanism may explain the broadening (slow exchange) of the smFRET histograms.

Is a binding of the B2-tract to the ordered region of the same or different pentameric unit sterically possible? Could such mechanism explain the apparent slow exchange in the FRET-efficiency histograms?

In the manuscript, Ficoll is considered as a crowding agent. Is it possible that Ficoll additionally acts as a seed for the formation of a liquid phase by providing an additional interface? If so, this should be discussed.

For NPMN118 the authors find no influence of the ionic strength on the Rg. Is this behavior expected given that NPM1N188 contains the B1 and A3 tract? Is it possible that an alternative mechanism describes these findings, e.g., an electrostatic interaction of the IDR with the folded subunit of the NPM1 pentamer (here the IDR of NPM1N188 may simply be too short to interact)? How do the authors exclude this alternative? Is this alternative sterically impossible?

The authors state that the Rg measured by SAXS for this construct corresponds to an ensemble of partially expanded conformations (Supplementary Fig. 2). The analysis of a SAXS curve is an underdetermined problem. Hence, it is unclear how the authors come to this conclusion. Do the

authors describe the SAXS-curve by a set of single structures or an ensemble of structures? How to the authors distinguish the two following scenarios? (1) An ensemble of partially expanded conformations. (2) An ensemble of extended and folded conformations.

The ensembles describing the SAXS curve should be represented. Which q -range was analyzed to obtain the R_g ? The experimental SAXS curves should be presented. Is there evidence of a liquid NPM1-phase in the SAXS-data at the given concentrations?

The smFRET measurements show similarly to their SAXS measurements a dependence on the ionic strength. The authors state that the FEH width is independent of the time binning. Does the width, expressed in DA-distances, increase or decrease with the salt concentration?

The heterogeneity in NPM1 (MW ~ 32 kDa) found by analytical ultracentrifugation is a direct evidence that NPM1 pentamers (MW ~ 160 kDa) and higher order oligomers. The authors find a population of $\sim 20\%$ of dimers of pentamers, reflected by the second peak in Fig.4a. However, neither the table heading of Supplementary Table 2 nor the table contents seem to correspond to the Figures. What are the N294 constructs mentioned in the table header? The percentage ($\sim 20\%$) of NPM1 dimers of pentamers for the NPM1WT (MW ~ 320 kDa) in the main text does not correspond to the percentages of the supplemental table. Are these values mass fractions?

Reviewer #4 (Remarks to the Author):

In this generally well-written paper, the authors draw on a series of biophysical studies to generate data in support of their model of liquid-liquid phase separation (LLPS) in the generation of membrane-less organelles (MLO) within the nucleolus. In particular, the paper focuses on the self-association of the protein NPM1 and its hetero-associations with a non-ribosomal protein, SURF-6 and ribosomal RNA. Changes to the structure of NPM1 could be induced by changes to the local salt concentration and were thus confirmed to be electrostatic in nature. Furthermore the authors demonstrate that formation of droplets indicative of MLOs in an NPM1-only system is mediated by electrostatic interactions between the intrinsically disordered regions (IDR) of the protein and that the same regions are important for hetero-associations with rRNA and SURF-6. The authors speculate that phosphorylation of Ser and Thr in the IDR changes the charge nature of this part of NPM1 and regulates its affinity towards different binding partners in LLPS.

The findings appear to be novel and concern an important area of research. This is not my area of research but I imagine that the findings will be of interest to others engaged in understanding the molecular mechanisms underlying the formation and dynamics of the nucleolar matrix. The work also serves as a nice example of the use of several complementary biophysical methods for probing a system. It follows on from work published by the same group last year in eLife which focussed on hetero-interactions of NPM1 with R-motifs and rRNA.

I believe the conclusions drawn are original to the work presented. The work is mostly convincing but I feel that the authors are trying to conclude too much from their data. It is hard for me to pinpoint exactly why I feel this is the case, however. I think the work supports many of the conclusions they draw but there are other plausible models that probably are not refuted by the data. I am not in the nucleosomal protein field but from what I can see, the literature is not vast and this paper does seem to move the field onwards.

The statistics used in conjunction with the turbidity assays and analytical ultracentrifugation appear to be appropriate and valid. I cannot judge the statistical analysis of the SAXS data since these are not

presented. I am not expert in FRET and so do not know how appropriate or valid this analysis is. I think a researcher could repeat the experiments but not all the analysis.

I would ask for clarification about the following:

Is there an atomic resolution structure for SURF-6? If not, could it be predicted? Are the R-motifs on its surface?

The sedimentation coefficient of CTD could have been computed from its high-resolution structure. Is it 1 S, as in Supp Table 2 and Fig 4d(ii) & (iii)?

Page 6: the SAXS data (i.e. scattering curves and their associated $p(r)$ and Guinier analyses should be presented in the supplementary data.

It is surprising that having obtained a huge amount of SAXS data, the authors have not attempted to model against the full scattering curves using e.g. EOM to represent an ensemble of flexible structures. This would have been a valuable and objective way to learn more about the changes in structure of NPM1 in the various regimes.

Page 15: Does the R_g determined by $p(r)$ agree with that determined by Guinier analysis?

Page 17: Why were only interference data collected for AUC? Why not absorbance data too? Were the proteins dialysed and was the reference solvent dialysate? If not, the data may not be reliable, since interference optics are sensitive to everything, whilst absorbance optics are more forgiving and detect only components that absorb at the selected wavelength.

Figure 2: the phase boundary in (c) is in a different place compared with (a) and (b). It should be in the same place. In the legend, were the data not fitted with a Gaussian model (as opposed to "to")? Panel (d) R_g is in Å, not A.

The authors should explain why the data in Figure 3c(ii) (top panel) are so noisy. Presumably this is because of the low partition coefficient of NMP1N188? The upper limit to the y-axis in this panel is missing.

Figure 5b: is the first [NaCl] really 105 mM or is it 100 mM?

Figure 6: I don't understand the relationship between the text in the legend and the figure. The legend tells us that the turbidity study was performed "at 20 μ M concentration of the indicated NPM1 construct". In (a) the study is homotypic – i.e. the NPM1 constructs self-associating. This is promoted by the increasing concentration of whichever NPM1 is being studied (as indicated on the y-axis). So the legend text is meaningless. In panel (b) I understand that now the concentration of the NPM1 is being kept at 20 μ M while the concentration of S6N is varied. But in panel (c) it appears that the concentration of rRNA is being kept constant at 100 mg/ml and the concentration of NPM1 is changing – i.e. at odds with the legend text.

Figure 7: the panels are labelled (a) to (e) but the legend describes (a) to (d) – the labels do not correspond – this needs fixing.

The reference journals are inconsistently titled (re capitals and abbreviations).

In supplementary material and elsewhere clarify what NPM1C54 is.

Supp Table 2: the superscripts a to f do not match the footnotes a to e. The unit of sedimentation coefficient is (S), not (Svedberg). Give the sequence molecular masses for comparison with the experimentally determined values.

Supp Fig 2: Rg cannot be determined to 2 decimal places in Å units. The experimental Rg is 44.5 Å.

Minor comments

Page 10: should read "wherein binding of the B2-tract (to rRNA)"?

Page 14: should read "Each turbidity assay was reproduced in triplicate".

Page 15: should read "...and refolded by dialysis, to promote one labelled monomer per pentameric ring" – it is an overstatement to say "ensure" since this is not demonstrated. In any case – how was this checked?

Page 15: should read "noise ratio across all three samples".

Page 20: should read "The green dotted line is a visual aid".

Throughout: "data were" not "data was"; "compared with", not "compared to"; "sedimentation coefficient" not "s-value".

Response to Reviewer's comments

Responses to Reviewer #1

Reviewer #1: "Page 3-Introduction, 3rd Paragraph, second sentence:"...depletion of any one of these proteins does not prevent assembly of the nucleolus...a. Any one of the proteins listed in the sentence above or any nucleolar protein? The authors mention that NPM1 interacts with over 130 proteins and that many of those are in the nucleolus and have multivalent R-motifs."

Author reply: *We thank the reviewer for their comment; we changed the text to read,*

"Interestingly, though, depletion of any one of these aforementioned proteins does not prevent assembly of the nucleolus, suggesting that multiple scaffolding mechanisms underlie nucleolar assembly and possibly other membrane-less bodies."

Reviewer #1: "b. Has it been shown that a deletion of a combination of these proteins prevents assembly of the nucleolus? c. If NPM1 knockdown does not prevent assembly of the nucleolus then is LLPS separation functionally crucial for processes like ribosome biogenesis or are there compensatory mechanisms?"

Author reply: *While knockdown of NPM1 does not prevent formation of nucleolar bodies, its depletion or mutation has effects on the function of the nucleolus, such as on the efficiency of ribosome biogenesis and nucleolar stress response, as described in (Amin, Matsunaga et al. 2008, Maggi, Kuchenruether et al. 2008). These references are cited in the manuscript, in the last paragraph on page 3.*

Reviewer #1: "Page 4-Introduction, 3rd Paragraph, eighth sentence: "...that inter-IDR interactions mediate homotypic LLPS by NPM1 under moderately crowded conditions."

a. What defines moderately crowded conditions?

b. When does this occur physiologically and what are the functional implications?"

Author reply: *We reworded the "[...] moderately crowded conditions." to "[...] crowded conditions, near physiological levels". Molecular crowding is ubiquitous in living cells, with reported values between 100 – 300 mg/mL and has been demonstrated to affect polypeptide chain compaction, enzyme kinetics, etc. (Ellis 2001). We discuss potential functional implications in the Discussion, on page 12.*

Reviewer #1: “Page 14-Methods, 3rd Paragraph third sentence: ”...Each turbidity assays was reproduced...”

a. “assays” should be changed to “assay””

Author reply: *We thank the reviewer for carefully reading our manuscript. We corrected the typo.*

Reviewer #1: “Figures 2/5/6: The authors used green circles to indicate, most likely conditions, when LLPS occurred. However, this is not clear and needs to be indicated in the figure legend.”

Author reply: *We included the following clarification in the figure legends:*

“The dotted purple line is a visual aid that represents the phase boundary for NPM1 between the mixed (grey circles) and demixed (green circles) states.”

Reviewer #1: “Figure 7: This is a nice figure, but it would be ideal to have an explanation of the possible functionality of the different types of interactions to show the biological relevance.”

Author reply: *We thank the reviewer for this suggestion. We have added panel Fig. 7f which illustrates a proposed mechanism of molecular hand-offs, in which NPM1 engages both rRNA and ribosomal proteins through heterotypic LLPS, mediating their interaction and association, with the NPM1 homotypic mechanism functioning as a back-up LLPS mechanism to retain NPM1 within the nucleolus while the assembled pre-ribosomal subunits exit the nucleolus as part of the ribosome biogenesis process. This model was explained in the Discussion section in the original form of the manuscript but not illustrated, as it is now.*

Reviewer #1: “6) Figure 3 ai/bi/ci: The scale bar is barely visible and could be enlarged.

7) The scale bar on some figures (Figure 3 ai/bi/ci, Figure 5b) are labeled with the value (10 μ m) but others are not (Figure 1C, Figure 5A). It would be ideal if they were all consistent.

8) There are two scale bars on Figure 1C (one vertical and one horizontal). This is not the case on any other figure and is redundant and could be removed.”

Author reply: *We thank the reviewer for these comments. We removed the vertical scale bars and adjusted the widths of all scale bars across all figures in the manuscript to maintain a consistent size.*

Responses to Reviewer #2

Reviewer #2: “The polymer phase separation field suggests that compaction often accompanies phase separation while there have been alternative suggestions within the protein phase separation community. Figure 7 depicts self-assembly in an extended conformation. Could the authors be more pedagogical in explaining how their data support this model or if there could be alternative interpretations?”

Author reply: *We recognize the validity of the Reviewer #2’s comment that a large body of literature suggests that crowding promotes polypeptide chain compaction. We altered the schematic representation to represent the NPM1 chains that are not involved in pentamer-pentamer crosslinks as being compact. Our data (SAXS, smFRET and NMR) support a model where NPM1-NPM1 crosslinks are mediated by electrostatic interactions between IDRs and that the IDR adopts a compact conformation in monodisperse pentamers. Collectively, while not unequivocally refuting alternative interpretations, these data strongly suggest that the IDR chain must expand from its intra-molecular compacted state, in order to interact in trans with other IDR chains.*

Responses to Reviewer #3

Reviewer #3: “It is unclear how the microscopic images were obtained. The images were probably taken on the surface of coverslips. If this is the case, does the NPM1 phase form in the liquid bulk phase or the glass water interface?”

Author reply: *While droplets form in solution, the confocal microscopic images shown in the manuscript are of droplets that have sedimented onto the coverslip surface. Droplets in solution above the coverslip surface are very mobile and quickly drift out of focus. We now include videos of NPM1^{WT}, NPM1^{N240} and NPM1^{N188} with SURF6-N forming droplets that “rain” from the solution onto the coverslip (Videos 1-3). We made a note of this technical detail in the Materials and Methods section, on page 15:*

“Droplets form in the bulk solution and settle due to gravity onto the surface of the coverslip, as shown in Videos 1-3. All reported images were recorded at the coverslip surface. Due to the rapid movement of the floating droplets, imaging and subsequent quantification of them was not possible, due to them drifting out of focus.”

Reviewer #3: “It seems that the fluorescence quantum yields were not accounted for in the analysis of the partition coefficients. Alexa647 and Alexa488 have significant different quantum yields, QY. Moreover, Alexa647 is an environment sensitive dye, i.e., viscosity dependent QY. If the presented analysis considered the QY and the detection efficiencies of the channels, the analysis should be described in more detail.”

Author reply: *The reviewer raises a valid point regarding the dependence of the quantum yields of AlexaFluor dyes on viscosity. In order to address this point, we performed fluorescence emission measurements of free AlexaFluor 488 and AlexaFluor 647 in water:glycerol mixtures of increasing viscosity, covering the 0.001 – 1.4 Pa·s range. We included a supporting figure showing raw emission spectra of the two dyes under the full range of viscosity solutions and their quantification (Supplementary Fig. 6a-c), as well as the following clarification in the Materials and Methods section:*

“Fluorescence spectroscopy. *“Aqueous stock solutions of Maleimide AlexaFluor 488-C5 and maleimide AlexaFluor 647-C2 fluorescent dyes were diluted 1:50 to 1 μ M into water:glycerol solutions at the specified percentage glycerol. Emission spectra were collected at 20 °C, using a QuantaMaster400 fluorometer (Horiba, Kyoto, Japan). AlexaFluor 488 samples were excited at 485 nm and emission data was recorded between 490-600 nm. AlexaFluor 647 samples were excited at 645 nm and emission data recorded between 650-800 nm (Supplementary Fig. 6a-b). In order to correlate the quantum yield changes observed by fluorescence spectroscopy with the microscopy data, we integrated the emission intensity measured spectroscopically over the wavelength range of the microscope bandpass filters: 525/50 BP (500-550 nm) and 725/150 (650-800 nm) for AlexaFluor 488 and AlexaFluor647, respectively. The integrated emission intensities were normalized with respect to the value for each AlexaFluor dye in water. Microrheology measurements of viscosity values within liquid-like and gel-like protein droplets indicated values between 0.7 Pa·s and 100 Pa·s (Elbaum-Garfinkle, Kim et al. 2015, Zhang, Elbaum-Garfinkle et al. 2015, Feric, Vaidya et al. 2016). Therefore, we defined the correction factor for each dye as the average of relative change values in integrated emission signal at $\eta > 0.3$ Pa·s, where the variation appears to plateau (Supplementary Fig. 6c).”*

Reviewer #3: “The experimental SAXS-curves, the model function, and the analyzed q-range should be presented in a figure including the weighted residuals of the models. (b) The analysis by SASSIE is nearly undocumented. It is unclear if the chi2 (supplementary Figure 2) is calculated for a single structure or an ensemble of structures. (c) If the chi2 corresponds to single structures out of an ensemble, the authors should consider weakening the statement that NPM1 adopts ensembles

of partly compact states, as a mixture of extended and collapsed structures may also describe the data.”

“The ensembles describing the SAXS curve should be represented. Which q-range was analyzed to obtain the R_g ? The experimental SAXS curves should be presented. Is there evidence of a liquid NPM1-phase in the SAXS-data at the given concentrations?”

Author reply: *We now graphically represent the q range utilized in the GNOM analysis of the scattering curves in Supplementary Fig. 2 and specify these values in the Materials and Methods, SAXS section (page 16):*

“The reduced data was analyzed with the ATSAS suit, using PRIMUS and GNOM for Guinier and $P(r)$ analyses, with a q range between $0.018 - 0.31 \text{ \AA}^{-1}$. Reported R_g values were obtained from the $P(r)$ analysis performed with GNOM, although similar values were obtained using the Guinier approximation performed with Primus.” Additionally, we now show representative graphs of the raw scattering data overlaid with the GNOM fit, as well as representative $P(r)$ plots, as a function of ionic strength (Supplementary Fig. 2).

We do not observe signs of phase separation in our SAXS data (significant upturn in the low q region or decrease in signal intensity due to dense phase settling) with any of the NPM1 constructs at $30 \mu\text{M}$ concentration.

The χ^2 values reported for the SASSIE calculations correspond to single structures. In order to address the reviewer’s concerns regarding the possibility for interpreting the convoluted scattering contribution of a mixture of highly compact and highly extended conformations vs. a rather homogeneous ensemble of partially extended conformations, we added the following clarification paragraph in the Materials and Methods section on page 16-17:

“We sampled the potential range of the conformational landscape with SASSIE to create representative conformers and then compared each conformer to the experimental curve to obtain a χ^2 value. The χ^2 vs. R_g plot (Supplementary Fig. 3) exhibits a broad basin near the experimentally observed R_g value of 44 \AA , suggesting that NPM1^{N188} dynamically exchanges between conformers with a range of compaction values. However, because the SAXS curves reflect only the ensemble averaged conformation, we cannot define minimum and maximum compaction limits for NPM1^{N188} molecules that experience conformational averaging.”

Additionally, we introduced the following explanation, in the main text, on page 6:

“The R_g value measured by SAXS for this construct, as well as the pair-wise distance distribution are constant over the entire range of salt concentrations tested (Fig. 2d and Supplementary Fig 2c), suggesting that NPM1^{N188} does not undergo ionic strength-dependent conformational changes. Molecular modeling based on SASSIE (Curtis, Raghunandan et al. 2012) (Supplementary Fig. 3), supports the hypothesis that NPM1^{N188} adopts an ensemble of partially expanded conformations, likely caused by electrostatic repulsion within the highly negatively charged, truncated IDR (Fig. 1a; estimated charge at pH 7.5, -37.0 (<http://protcalc.sourceforge.net/>); Supplementary Table 1).”

Reviewer #3: “When describing the smFRET setup the authors refer to Ferreon et al. 2009 where ALEX-excitation was used. In the manuscript solely the donor-excitation is mentioned. Did the authors employ ALEX or PIE? If this is the case, does the variance of acceptor brightness distribution explain the broad FRET efficiency histogram? Do the authors select FRET molecules by the directly excited acceptor? If so, which selection criteria were used?”

Author reply: *For smFRET, we used single excitation for these experiments, as for most of the experiments in the Ferreon et al. 2009 paper that the reviewer has noted and we cited. Our smFRET data trends with salt are consistent with and support the SAXS results. Furthermore, we also add that we and others have used the AlexaFluor 488/594 dye pair on several other protein systems without observing such broadening, hence the effect is not dye specific. We have a more detailed discussion of the interpretation in the Supplementary Note 1.*

Reviewer #3: “The recording of the FRAP curves is unclear. Which region of interest, ROI, was bleached? An entire NPM1 droplet or parts a droplet? Together with the FRAP curves the ROI before bleaching, after bleaching, and the recovered region could be shown.”

Author reply: *We included the following account detailing the FRAP experiments in the Materials and Methods section, on page 15:*

“For the FRAP experiments, a circular region of interest (ROI) of 1 μm in diameter, located at the center of the droplets was photobleached by illuminating the ROI with the laser set at 100% power for 1 ms.”

Additional images were included in Supplementary Fig. 6d, to illustrate the droplets immediately after photobleaching ($t = 0$ s) and 50 s after bleaching.

Reviewer #3: “Analytical ultracentrifugation) Figure 4 may be simplified to stress the statement that the sample is heterogeneous. The main finding is that NPM1WT forms dimers of pentamers and that NPM1N240 is heterogeneous. The supplementary table 2 seems not to coincide to Fig. 4 of the main text.”

“The superscripts of the header of supplementary table 2 are wrong. Please also give the protein NPM1 concentration in mM for easier comparison with the other techniques.”

“The heterogeneity in NPM1 (MW ~32kDa) found by analytical ultracentrifugation is a direct evidence that NPM1 pentamers (MW ~160 kDa) and higher order oligomers. The authors find a population of ~ 20% of dimers of pentamers, reflected by the second peak in Fig.4a. However, neither the table heading of Supplementary Table 2 nor the table contents seem to correspond to the Figures. What are the N294 constructs mentioned in the table header? The percentage (~20%) of NPM1 dimers of pentamers for the NPM1WT (MW ~ 320 kDa) in the main text does not correspond to the percentages of the supplemental table. Are these values mass fractions?”

Author reply: *We thank the reviewer for pointing out the inconsistency between the analyses shown in the table and the figure. The table included in the first submission of the manuscript corresponded to the 1 D analysis of the AUC data. We replaced the table with the correct analysis of the 2D SV-AUC data shown in Fig. 4. We now express the protein concentrations in μM instead of mg/mL, as suggested by the reviewer. We simplified the presentation of AUC data by removing the 1 D sedimentation velocity plots from the Supplementary figures. This does not affect the conclusions drawn from the AUC data.*

Reviewer #3: “The abbreviation SURF6, Surfeit locus protein 6, is not introduced.”

Author reply: *We introduced the abbreviation on page 5, first paragraph.*

Reviewer #3: “The naming of SURF6-N is inconsistent, in the figures SURF6-N is sometimes addressed by SN6 (Fig. 6)”

Author reply: *We corrected the label in Fig. 6.*

Reviewer #3: “The abbreviation A-tracts is used before being defined.”

Author reply: *We defined A-tracts on pg. 3, last paragraph.*

Reviewer #3: “The list of mutants (Fig. S1) is useful and thus may be presented in the main text

Fig.1.”

Author reply: *Per reviewer’s suggestion, we moved the schematic representation of the mutants used in this study from Supplementary Fig. 1 to main Fig. 1d.*

Reviewer #3: “Figures are referred without discussion of their content, e.g., Fig. 1a.”

“Most figures are insufficiently described by their legends, e.g., in the legend of Fig. 1c it is not mentioned that microscopic images are shown.”

Author reply: We revised the text to explicitly refer Fig. 1a. On page 4, first paragraph, the text reads “Fig. 1a-b illustrate the clustered charges and the sub-domain organization in NPM1.” Additionally, we revised the legend of Fig. 1c to read “**(c)** Confocal microscopy images of phase separation by 20 μ M NPM1^{WT} with 20 μ M SURF6-N in buffer with variable concentrations of NaCl, as indicated.”

Reviewer #3: “Fig. 1c. AlexaFluor647 instead of AlexaFluor640 - I guess”

Author reply: *The reviewer is correct. We replaced “AlexaFluor640” with “AlexaFluor647” in Fig. 1c legend.*

Reviewer #3: “For NPMN118 the authors find no influence of the ionic strength on the R_g . Is this behavior expected given that NPM1N188 contains the B1 and A3 tract? Is it possible that an alternative mechanism describes these findings, e.g., an electrostatic interaction of the IDR with the folded subunit of the NPM1 pentamer (here the IDR of NPM1N188 may simply be too short to interact)? How do the authors exclude this alternative? Is this alternative sterically impossible?” “Is it also possible, that the IDR is either bound to the folded pentameric unit of NPM1 or “loose”? Such mechanism may explain the broadening (slow exchange) of the smFRET histograms.” “Is a binding of the B2-tract to the ordered region of the same or different pentameric unit sterically possible? Could such mechanism explain the apparent slow exchange in the FRET-efficiency histograms?”

Author reply: *The truncated IDR is a negatively charged strong polyelectrolyte (calculated charge - 30.2; Supplementary Table 1) and is predicted to adopt a swollen coil conformation according to CIDER (<http://pappulab.wustl.edu/CIDER/analysis/>):*

Based on the SASSIE calculations (see Supplementary Fig. 3, $R_g = 32 \text{ \AA}$ conformation), compact conformations where the truncated IDR interacts with the folded oligomerization domain are sterically possible, but incur a high energetic penalty.

As it pertains to the smFRET data, if the B2-tract is binding to the ordered region of the pentameric unit, it could lead to slower exchange in the FRET histograms. Given the

smFRET data, it is not possible to exclude it from the alternative possibilities discussed in Supplementary Note 1. In order to discriminate between the mechanism that we emphasize in the paper and the plausible alternative scenario suggested by the reviewer, we performed NMR experiments ($2D \text{ } ^1\text{H-}^{15}\text{N}$ HSQC experiments and $1D \text{ } ^{15}\text{N}$ -filtered ^1H diffusion experiments), presented in the revised manuscript as Supplementary Fig. 4. Collectively, these new results show that IDR domains interact weakly in trans in a homotypic manner, while no significant interactions can be detected at the same molar concentrations with the folded domains. The following paragraph is now included in the Results section, on page 7:

“The loss of conformational sensitivity to ionic strength upon deletion of the B2-tract and CTD in $\text{NPM1}^{\text{N188}}$ could be explained through three distinct mechanisms which could cause IDR compaction: (1) B2-tract interacts with the A-tracts, (2) B2-tract interacts with the OD, and (3) IDR interacts with CTD. In order to discriminate between these three mechanisms, we performed two-dimensional $^1\text{H}/^{15}\text{N}$ HSQC (Supplementary Fig. 4a) and $1D \text{ } ^{15}\text{N}$ -filtered ^1H diffusion (Supplementary

Fig. 4b-c & Table 3) experiments with 30 μM ^{15}N NPM1^{IDR} in the presence or absence of excess, non-isotope-labeled NPM1^{IDR}, NPM1^{OD} and NPM1^{CTD} (see Fig. 1b). Small chemical shift perturbations and slowed diffusion, indicative of weak interactions, were observed for ^{15}N NPM1^{IDR} in the presence of excess NPM1^{IDR}, but not either of the folded domains (Supplementary Fig. 4). Thus, the NMR analysis supports a model wherein interactions between the B2-tract and the A-tracts within the IDR are responsible for the ionic strength-dependent conformational changes in NPM1^{WT} and NPM1^{N240}.”

Reviewer #3: “In the manuscript, Ficoll is considered as a crowding agent. Is it possible that Ficoll additionally acts as a seed for the formation of a liquid phase by providing an additional interface? If so, this should be discussed.”

Author reply: *While the possibility that Ficoll may act as a seed for the formation of the liquid phase cannot be completely ruled out, we have unpublished results that show that homotypic phase separation of NPM1 occurs in the presence of different types of crowding agents, including PEG of various molecular sizes and Ficoll. Thus, we argue that the principal driver for phase separation is in fact excluded volume caused by molecular crowding, rather than seeding. However, for simplicity, we prefer to not enter into this discussion in the manuscript.*

Reviewer #3: “The authors state that the R_g measured by SAXS for this construct corresponds to an ensemble of partially expanded conformations (Supplementary Fig. 2). The analysis of a SAXS curve is an underdetermined problem. Hence, it is unclear how the authors come to this conclusion. Do the authors describe the SAXS-curve by a set of single structures or an ensemble of structures? How to the authors distinguish the two following scenarios? (1) An ensemble of partially expanded conformations. (2) An ensemble of extended and folded conformations.”

Author reply: *To address this point, we added the following clarification in the Materials and Methods section, on page 16:*

“We sampled the potential range of the conformational landscape with SASSIE to create representative conformers and then compared each conformer to the experimental curve to obtain a χ^2 value. The χ^2 vs. R_g plot (Supplementary Fig. 3) exhibits a broad basin near the experimentally observed R_g value of 44 Å, suggesting that NPM1^{N188} dynamically exchanges between conformers with a range of compaction values. However, because the SAXS curves reflect only the ensemble averaged conformation, we cannot define minimum and maximum compaction limits for NPM1^{N188} molecules that experience conformational averaging”

Reviewer #3: “The smFRET measurements show similarly to their SAXS measurements a dependence on the ionic strength. The authors state that the FEH width is independent of the time binning. Does the width, expressed in DA-distances, increase or decrease with the salt concentration?”

Author reply: *Peak broadening was present in the system regardless of integration time, where we went as low as 200 μ s (Supplementary Note 1). Generally, the width increases as a function of salt concentration in terms of DA-distances, which is consistent with the SAXS results.*

Responses to Reviewer #4

Reviewer #4: “Is there an atomic resolution structure for SURF-6? If not, could it be predicted? Are the R-motifs on its surface?”

Author reply: *SURF6 is an intrinsically disordered protein and there are no atomic structures of it. Supplementary Fig. 1e (bottom panel) illustrates the disorder prediction results for this protein. In the revised version of the manuscript we now include a 2D $^1\text{H}/^{15}\text{N}$ HSQC NMR spectrum of SURF6-N (Supplementary Fig. 1c) which shows poor chemical shift dispersion in the ^1H dimension, an indication of structural disorder. Therefore, we expect that the R-motifs will be readily accessible for interactions with its binding partners.*

Reviewer #4: “Page 6: the SAXS data (i.e. scattering curves and their associated $p(r)$ and Guinier analyses should be presented in the supplementary data.”

“Page 15: Does the R_g determined by $p(r)$ agree with that determined by Guinier analysis?”

Author reply: *We now graphically represent the q range utilized in the GNOM analysis of the scattering curves in Supplementary Fig. 2 and specify these values in the Materials and Methods, SAXS section (page 16):*

“The reduced data was analyzed with the ATSAS suit, using PRIMUS and GNOM for Guinier and $P(r)$ analyses, with a q range between $0.018 - 0.31 \text{ \AA}^{-1}$. Reported R_g values were obtained from the $P(r)$ analysis performed with GNOM, although similar values were obtained using the Guinier approximation performed with Primus.” Additionally, we now show representative graphs of the raw scattering data overlaid with the GNOM fit, as well as representative $P(r)$ plots, as a function of ionic strength (Supplementary Fig. 2).

Reviewer #4: “It is surprising that having obtained a huge amount of SAXS data, the authors have not attempted to model against the full scattering curves using e.g. EOM to represent an ensemble of flexible structures. This would have been a valuable and objective way to learn more about the changes in structure of NPM1 in the various regimes.”

Author reply: *This is a good suggestion (to use EOM to extract additional insights from the SAXS data). However, our initial attempts to calculate ensembles for the 3 different constructs (NPM1^{N188}, NPM1^{N240}, NPM1^{WT}) based on SAXS data did not yield satisfactory results. So far, with the EOM*

default settings and with increasing the pool size to 50k, we have been unable to obtain good fits to the data, which likely indicates that we need to generate even larger pools. Considering the time involved, and that we have already obtained strong structural information from the current analysis of the SAXS data, we believe it is appropriate to exclude the EOS analysis from the current manuscript. This being said, we will continue to work with EOM v2.0 to hopefully be able to generate ensemble models for NPM1 based on SAXS data in the future. We further note that we have successfully used EOM v1.0 in the past with different flexible systems, but the newer EOM v2.0 is needed here for the pentamer structure of NPM1 and it is not providing reasonable results thus far.

Reviewer #4: “Page 17: Why were only interference data collected for AUC? Why not absorbance data too? Were the proteins dialysed and was the reference solvent dialysate? If not, the data may not be reliable, since interference optics are sensitive to everything, whilst absorbance optics are more forgiving and detect only components that absorb at the selected wavelength.”

Author reply: *We prefer to use Raleigh Interference optics as a first option of detection, due to its improved sensitivity over the absorbance optics and more rapid (~ 20-fold) data sampling during collection. Additionally, the absorbance is limiting since an OD of larger than 1-1.2 will saturate the optics, while interference optics has a much higher upper limit for protein concentration. In addition, due to the deletion of the two tryptophans in the C-terminus on NPM1^{WT}, the molar extinction coefficients of the NPM1^{N240} and NPM1^{N188} are low, and thus, they are poorly suited for absorbance optics. Indeed, all samples were buffer exchanged into the reference buffer by overnight dialysis, thereby the buffer mismatch is minimized. This technical aspect is now detailed in the Materials and Methods section on page 18:*

“The samples, dialyzed overnight against the reference buffer (10 mM Tris pH 7.5, 150 mM NaCl, and 2 mM DTT) [...]”.

Reviewer #4: “The authors should explain why the data in Figure 3c(ii) (top panel) are so noisy. Presumably this is because of the low partition coefficient of NMP1N188? The upper limit to the y-axis in this panel is missing.”

Author reply: *The reviewer is correct, that the FRAP data for NPM1^{N188}, shown in panel 3c (ii) is noisy due to the low partition coefficient, which in effect translates into low signal-to-noise ratio.*

Reviewer #4: “Figure 5b: is the first [NaCl] really 105 mM or is it 100 mM?”

Author reply: Yes, the 105 mM NaCl concentration is correct.

Reviewer #4: “Figure 6: I don’t understand the relationship between the text in the legend and the figure. The legend tells us that the turbidity study was performed “at 20 μ M concentration of the indicated NPM1 construct”. In (a) the study is homotypic – i.e. the NPM1 constructs self-associating. This is promoted by the increasing concentration of whichever NPM1 is being studied (as indicated on the y-axis). So the legend text is meaningless. In panel (b) I understand that now the concentration of the NPM1 is being kept at 20 μ M while the concentration of S6N is varied. But in panel (c) it appears that the concentration of rRNA is being kept constant at 100 mg/ml and the concentration of NPM1 is changing – i.e. at odds with the legend text.”

Author reply: We clarified the statement in legend. The text now reads:

“Phase separation diagrams based on turbidity assays for the homotypic (a), heterotypic with SURF6-N (NPM1 constructs at 20 μ M) (b) and heterotypic with rRNA (c) mechanisms, at the indicated NPM1 construct (a, c) or SURF6-N (b) concentrations. The purple dotted line is a visual aid that represents the phase boundary for NPM1^{WT} between the mixed (grey circles) and demixed (green circles) states.”

Reviewer #4: “Figure 7: the panels are labelled (a) to (e) but the legend describes (a) to (d) – the labels do not correspond – this need fixing.”

Author reply: We rectified the legend to correctly correspond with the figure panel labeling.

Reviewer #4: “In supplementary material and elsewhere clarify what NPM1C54 is.”

Author reply: We define NPM1C54 (now NPM1^{CTD}) on page 8. Since this construct encompasses the C-terminal domain of NPM1, we adopted the more intuitive nomenclature of NPM1^{CTD}. Additionally, we now show a schematic representation of this construct in Fig. 1b.

Reviewer #4: “Supp Table 2: the superscripts a to f do not match the footnotes a to e.”

Author reply: We thank the reviewer for pointing out the mismatch between the labels and footnotes. The table included in the first submission of the manuscript corresponded to the 1 D analysis of the AUC data. We replaced the table with the correct analysis of the 2D SV-AUC data

shown in Fig. 4. We also simplified the presentation of AUC data by removing the 1 D sedimentation velocity plots from the Supplementary figures. This does not affect the conclusions drawn from the AUC data.

Reviewer #4: “Figure 2: ... In the legend, were the data not fitted with a Gaussian model (as opposed to "to")? Panel (d) R_g is in Å, nor A.”

“Supp Fig 2: R_g cannot be determined to 2 decimal places in Å units. The experimental R_g is 44.5 Å.”

“Page 10: should read "wherein binding of the B2-tract (to rRNA)"?”

“Page 14: should read "Each turbidity assay was reproduced in triplicate".”

“Page 15: should read "...and refolded by dialysis, to promote one labelled monomer per pentameric ring" – it is an overstatement to say "ensure" since this is not demonstrated. In any case – how was this checked?”

“Page 15: should read "noise ratio across all three samples".”

“Page 20: should read "The green dotted line is a visual aid".”

“Throughout: "data were" not "data was"; "compared with", not "compared to"; "sedimentation coefficient" not "s-value".”

Author reply: *We thank the reviewer for the careful revision of our manuscript. Throughout the manuscript, we corrected the typos and the inconsistencies in figures and figure legends, pointed out by the Reviewer #4.*

Reviewer #4: “Figure 2: the phase boundary in (c) is in a different place compared with (a) and (b). It should be in the same place.

Author reply: *We do not understand this comment. A key point in comparison of Figs. 2a, b and c is that the concentrations for heterotypic LLPS by NPM1^{N188} are lower than for the other two constructs, suggesting that the B2 tract (in NPM1^{N240}) and the B2 tract and CTD (in NPM1^{WT}) limit the accessibility of the A-tracts for interactions with the multiple Arg-rich motifs (R-motifs) in SURF6-N. While NPM1^{N188} is “better” at heterotypic LLPS, it is unable to undergo homotypic LLPS. These results, together, illustrate the competitive interplay between NPM1’s heterotypic and homotypic mechanisms of LLPS. The differences between the noted figure panels were critical in deciphering these competitive mechanisms. We hope this clarifies these data and their significance for the Reviewer.*

References for this discussion with reviewers

- Amin, M. A., S. Matsunaga, S. Uchiyama and K. Fukui (2008). "Depletion of nucleophosmin leads to distortion of nucleolar and nuclear structures in HeLa cells." *Biochem J* **415**(3): 345-351.
- Curtis, J. E., S. Raghunandan, H. Nanda and S. Krueger (2012). "SASSIE: A program to study intrinsically disordered biological molecules and macromolecular ensembles using experimental scattering restraints." *Computer Physics Communications* **183**(2): 382-389.
- Elbaum-Garfinkle, S., Y. Kim, K. Szczepaniak, C. C. Chen, C. R. Eckmann, S. Myong and C. P. Brangwynne (2015). "The disordered P granule protein LAF-1 drives phase separation into droplets with tunable viscosity and dynamics." *Proc Natl Acad Sci U S A* **112**(23): 7189-7194.
- Feric, M., N. Vaidya, T. S. Harmon, D. M. Mitrea, L. Zhu, T. M. Richardson, R. W. Kriwacki, R. V. Pappu and C. P. Brangwynne (2016). "Coexisting Liquid Phases Underlie Nucleolar Subcompartments." *Cell* **165**(7): 1686-1697.
<http://protcalc.sourceforge.net/>
- Maggi, L. B., Jr., M. Kuchenruether, D. Y. Dadey, R. M. Schwoppe, S. Grisendi, R. R. Townsend, P. P. Pandolfi and J. D. Weber (2008). "Nucleophosmin serves as a rate-limiting nuclear export chaperone for the Mammalian ribosome." *Mol Cell Biol* **28**(23): 7050-7065.
- Zhang, H., S. Elbaum-Garfinkle, E. M. Langdon, N. Taylor, P. Occhipinti, A. A. Bridges, C. P. Brangwynne and A. S. Gladfelter (2015). "RNA Controls PolyQ Protein Phase Transitions." *Mol Cell* **60**(2): 220-230.

REVIEWERS' COMMENTS:

Reviewer #1 (Remarks to the Author):

The authors answered all critical points and revised manuscript accordingly.

Reviewer #3 (Remarks to the Author):

This highly interesting study unequivocally demonstrates how the self-interaction of NPM1 modulates mechanisms of liquid-liquid phase separations. The work has been designed well, performed very carefully and written up at a high level of scholarship. The revised manuscript is clear and concise, yet it contains all necessary details. Moreover, the manuscript addressed all scientific concerns raised in the first round of the peer review.

Distribution of the material between the article and Supporting Information is well thought. All conclusions are well supported by experimental evidence and uncertainties are clearly marked and discussed.

This is a fine piece of work I recommend to be published in Nature Communications in its present form. Editorial changes and corrections could be done at proof stage.

Thomas Peulen